# GOMAA-Geo: GOal Modality Agnostic Active Geo-localization

**Anindya Sarkar**[1][*]**, Srikumar Sastry**[1][*]**, Aleksis Pirinen**[2,3]**,**
**Chongjie Zhang**[1]**, Nathan Jacobs**[1]**, Yevgeniy Vorobeychik**[1]
[1]Department of Computer Science and Engineering, Washington University in St. Louis
[2]RISE Research Institutes of Sweden
[3]Swedish Centre for Impacts of Climate Extremes (climes)
Corresponding authors: {anindya, s.sastry}@wustl.edu

## Abstract

We consider the task of *active geo-localization* (AGL) in which an agent uses a sequence of visual cues observed during aerial navigation to find a target specified through multiple possible modalities. This could emulate a UAV involved in a search-and-rescue operation navigating through an area, observing a stream of aerial images as it goes. The AGL task is associated with two important challenges. Firstly, an agent must deal with a goal specification in one of multiple modalities (e.g., through a natural language description) while the search cues are provided in other modalities (aerial imagery). The second challenge is limited localization time (e.g., limited battery life, urgency) so that the goal must be localized as efficiently as possible, i.e. the agent must effectively leverage its sequentially observed aerial views when searching for the goal. To address these challenges, we propose *GOMAA-Geo* – a goal modality agnostic active geo-localization agent – for zero-shot generalization between different goal modalities. Our approach combines cross-modality contrastive learning to align representations across modalities with supervised foundation model pretraining and reinforcement learning to obtain highly effective navigation and localization policies. Through extensive evaluations, we show that *GOMAA-Geo* outperforms alternative learnable approaches and that it generalizes across datasets – e.g., to disaster-hit areas without seeing a single disaster scenario during training – and goal modalities – e.g., to ground-level imagery or textual descriptions, despite only being trained with goals specified as aerial views. Our code is available at: https://github.com/mvrl/GOMAA-Geo.

## 1 Introduction

A common objective among many search-and-rescue (SAR) operations is to locate missing individuals within a defined search area, such as a specific neighborhood. To this end, one may leverage indirect information about the location of these individuals that may come from natural language descriptions, photographs, etc, e.g. through social media or eyewitness accounts. However, such information may not allow us to precisely identify actual locations (for example, these may not be provided on social media), and potential GPS information may be unreliable. In such situations, deploying a UAV to explore the area from an aerial perspective can be effective for accurate localization and subsequent assistance to those who are missing. However, the field of view of a UAV is typically limited (at least in comparison with the area to be explored), and the inherent urgency of search-and-rescue tasks imposes an effective temporal budget constraint. We refer to this general task and modeling framework as *goal modality agnostic active geo-localization*. Pirinen et al. [24] recently introduced a

---

[*]Equal contribution.

38th Conference on Neural Information Processing Systems (NeurIPS 2024).

deep reinforcement learning (DRL) approach for a significantly simpler version of this setup, where goals are always assumed to be specified as aerial images. This is severely limiting in practical scenarios, where goal contents are instead often available as ground-level imagery or natural language text (e.g., on social media following a disaster). Also, since no two geo-localization scenarios are alike, zero-shot generalizability is a crucial aspect to consider when developing methods for this task.

To this end, we introduce *GOMAA-Geo*, a novel framework for tackling the proposed GOal Modality Agnostic Active Geo-localization task. Our framework allows for goals to be specified in several modalities, such as text or ground-level images, whereas search cues are provided as a sequence of aerial images (akin to a UAV with a downwards-facing camera). Furthermore, *GOMAA-Geo* effectively leverages past search information in deciding where to search next. To address the potential misalignment between goal specification and observational modalities, as well as facilitating zero-shot transfer, we develop a cross-modality contrastive learning approach that aligns representation across modalities. We then combine this representation learning with foundation model pretraining and DRL to obtain policies that efficiently localize goals in a specification-agnostic way.

Given the scarcity of high-quality datasets that combine aerial imagery with other modalities like natural language text or ground-level imagery, evaluating *GOMAA-Geo* becomes challenging. To address this limitation, we have created a novel dataset that allows for benchmarking of multi-modal geo-localization. We then demonstrate that our proposed *GOMAA-Geo* framework is highly effective at performing active geo-localization tasks across diverse goal modalities, despite being *trained exclusively on data from a single goal modality* (i.e., aerial views).

In summary, we make the following contributions:

- We introduce *GOMAA-Geo*, a novel framework for effectively tackling goal modality agnostic active geo-localization – even when the policy is trained exclusively on data from a single goal modality – and perform extensive experiments on two publicly available aerial image datasets, which demonstrate that *GOMAA-Geo* significantly outperforms alternative approaches.
- We create a novel dataset to assess various approaches for active geo-localization across three different goal modalities: aerial images, ground-level images, and natural language text.
- We demonstrate the significance of each component within our proposed *GOMAA-Geo* framework through a comprehensive series of quantitative and qualitative ablation analyses.
- We assess the zero-shot generalizability of *GOMAA-Geo* by mimicking a real-world disaster scenario, where goals are presented as pre-disaster images and where policy training is done exclusively on such pre-disaster data, while the active geo-localization only has access to post-disaster aerial image glimpses during inference.

## 2   Related Work

**Geo-localization.** There is extensive prior work on one-shot *visual* geo-localization [41, 38, 46, 43, 26, 40, 34, 6, 5, 47, 12]. Such works aim to infer relationships between images from different perspectives, typically predicting an aerial view corresponding to a ground-level image. This problem is commonly tackled by exhaustively comparing a query image with respect to a large reference dataset of aerial imagery. Alternatively, as in [3], a model is trained to directly predict the global geo-coordinates of a given query image. In contrast, our active geo-localization (AGL) setup aims to localize a target from its location description in one of several modalities by *navigating* the geographical area containing it.

**LLMs for RL.** LLMs have been applied to RL and robotics for planning [35, 42, 15]. Our work instead focuses on learning goal-modality agnostic zero-shot generalizable agents and aims to leverage LLMs in order to learn goal-conditioned history-aware representations to guide RL agents for AGL.

**Embodied learning.** Our setup is also related to embodied goal navigation [2, 48, 19], where an agent should navigate (in a first-person perspective) in a 3d environment towards a goal specified e.g. as an image. This task may sometimes be more challenging than our setup since the agent needs to explore among obstacles (e.g. walls and furniture). On the other hand, the agent may often observe the goal from far away (e.g. from the opposite side of a recently entered room), while our task is more challenging in that the goal can never be partially observed prior to reaching it.

**Autonomous UAVs.** There are many prior works about autonomously controlling UAVs [36, 18, 10, 4, 30, 45, 25]. Many of these (e.g. [36, 30, 45]) revolve around exploring large environments efficiently,

so that certain inferences can be accurately performed given only a sparse selection of high-resolution observations. There are also works that are closer to us in terms of task setup [4, 18, 10, 32, 31, 24]. For example, [24] also considers the task of actively localizing a goal, assuming an agent with aerial view observations of a scene. However, this work only considers the idealized setting in which the goal is specified in terms of a top-view observation from the exact same scenario in which the agent operates. In contrast, we allow for flexibly specifying goals in an agnostic manner.

**Multi-modal representation learning in remote sensing.** Recent studies [17, 11] have shown that satellite image representations can be aligned with the shared embedding space learned by CLIP [28], by using co-located ground-level imagery as an intermediary to link satellite images and language. We utilize such aligned multi-modal embedding to represent goals in a modality-agnostic manner.

## 3 Active Geo-localization Setup

To formalize the active geo-localization (AGL) setup, we consider an agent (e.g. a UAV in a search-and-rescue scenario) that aims to localize a goal within a pre-defined search area. This area is discretized into a $X \times Y$ grid superimposed on a given aerial landscape (larger image), with each grid corresponding to a position (location) and representing the limited field of view of the agent (UAV) – i.e., the agent can only observe the aerial content of a sub-image $x_t$ corresponding to the grid cell in which it is located at time step $t$. The agent can move between cells by taking actions $a \in \mathcal{A}$ (up, down, left, right, in a canonical birds-eye-view orientation).

The search goal is associated with one of the grid cells, including a description in one of several modalities, and the agent's task is to reach this goal – relying solely on visual cues in the form of sequentially observed aerial sub-images – within a fixed number of steps $\mathcal{B}$. More precisely, let $s_g$ be the semantic content and $p_g$ the location (within the search area) of the goal. We use $x_g$ to denote the provided *description* of the goal, available to the agent in the form of either **(i)** *natural language text*, **(ii)** *ground-level image*, or **(iii)** *aerial image* (sub-image within the top-view perspective search area). Notably, the true location $p_g$ of the goal is *not* provided to the agent, so it must be inferred by the agent during the search process. The AGL task is deemed accomplished when the agent's current position $p_t$ aligns with the goal position $p_g$, i.e., when $p_t = p_g$. An overview is provided in Figure 1.

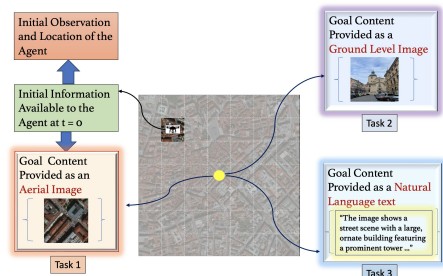

Figure 1: **Active geo-localization across different goal modalities.** The agent must navigate to the goal (yellow dot) based on *partial aerial glimpses*, i.e. the full area is never observed in its entirety.

We model this problem as a *Goal-Conditioned Partially Observable Markov Decision Process (GC-POMDP)* and consider a family of *GC-POMDP* environments $\mathcal{M}^e = \{(\mathcal{S}^e, \mathcal{A}, \mathcal{X}^e, \mathcal{T}^e, \mathcal{G}^e, \gamma) | e \in \epsilon\}$ where $e$ is the environment index. Each environment comprises a state space $\mathcal{S}^e$, shared action space $\mathcal{A}$, observation space $\mathcal{X}^e$, transition dynamics $\mathcal{T}^e$, goal space $\mathcal{G}^e \subset \mathcal{S}^e$, and discount factor $\gamma \in [0, 1]$. The observation $x^e \in \mathcal{X}^e$ is determined by state $s^e \in \mathcal{S}^e$ and the unknown environmental factor $b^e \in \mathcal{F}^e$, i.e., $x^e(s^e, b^e)$, where $\mathcal{F}^e$ encompasses variations related e.g. to disasters, diverse geospatial regions, varying seasons, and so on. We use $x_t^e$ to denote the observation at state $s^e$ and step $t$, for domain $e$.

The primary objective in a *GC-POMDP* is to learn a history-aware goal conditioned policy $\pi(a_t | x_{h_t}^e, g^e)$, where $x_{h_t}^e = (x_t^e, a_{t-1}, x_{t-1}^e, \ldots, a_0, x_0^e)$ combines all the previous observations and actions up to time $t$, that maximizes the discounted state density function $J(\pi)$ across all domains $e \in \epsilon$ as follows:

$$J(\pi) = \mathbb{E}_{e \sim \epsilon, g^e \sim \mathcal{G}^e, \pi} \left[ (1 - \gamma) \sum_{t=0}^{\infty} \gamma^t p_\pi^e(s_t = g^e | g^e) \right] \tag{1}$$

Here $p_\pi^e(s_t = g^e | g^e)$ represents the probability of reaching the goal $g^e$ at step $t$ within domain $e$ under the policy $\pi(.|x_{h_t}^e, g^e)$, and $e \sim \epsilon$ and $g \sim \mathcal{G}^e$ refer to uniform samples from each set. Throughout the training process, the agent is exposed to a set of training environments $\{e_i\}_{i=1}^N = \epsilon_{train} \subset \epsilon$, each identified by its environment index. We also assume during training that the goal content is

always available to the agent in the form of an *aerial image*. Our objective is to train a history-aware goal-conditioned RL agent capable of generalizing across goal modalities (such as natural language text, ground-level images, aerial images) and environmental variations, such as natural disasters.

# 4 Proposed Framework for Goal Modality Agnostic Active Geo-localization

In this section we introduce ***GOMAA-Geo***, a novel learning framework designed to address the goal modality agnostic active geo-localization (AGL) problem. *GOMAA-Geo* consists of three components: **(i)** *representation alignment across modalities*; **(ii)** *RL-aligned representation learning using goal aware supervised pretraining of LLM*; and **(iii)** *planning*. We next describe in detail each of these components within the proposed framework, and then we explain how these modules are integrated to train a goal-conditioned policy $\pi$, capable of generalizing to unseen test environments and unobserved goal modalities after training on $\epsilon_{train}$ (cf. Section 3).

**Aligning representations across modalities.** As we aspire to learn a goal modality agnostic policy $\pi$, it is essential to ensure that the embedding of the goal content – regardless of its modality (such as natural language text) – is aligned with the aerial image modality, as in the AGL setup we assume access only to aerial view glimpses during navigation. To this end, we take motivation from CLIP [28], which is designed to understand ground-level images and text jointly by aligning them in a shared embedding space through contrastive learning. Recent works [17, 11] have demonstrated that it is possible to align the representations of aerial images with the shared embedding space learned via CLIP. The key insight is to use co-located internet imagery taken on the ground as an intermediary for connecting aerial images and language. Following [17, 11], we proceed to train an image encoder tailored for aerial images, aiming to align it with CLIP's image encoder by utilizing a large-scale dataset comprising paired ground-level and aerial images. To align the embeddings of aerial images with those of ground-level images from CLIP, we employ contrastive learning on the aerial image encoder $s_\theta$ (parameterized by $\theta$) using the InfoNCE loss [21] in the following manner:

$$\mathcal{L}^{\text{align}} = \frac{1}{N} \sum_{i=0}^{i=N} -\log \left( \frac{\exp(s_\theta^i \cdot f_\phi^i / \tau)}{\sum_{j=0}^{j=N} \exp(s_\theta^i \cdot f_\phi^j / \tau)} \right) \tag{2}$$

Here we represent the CLIP image encoder as $f_\phi$ and $\tau$ denotes the temperature. We optimize the $\mathcal{L}^{\text{align}}$ loss (2) to reduce the gap between co-located aerial and ground-level images within the CLIP embedding space. It is important to highlight that the CLIP image encoder remains unchanged throughout training. Therefore, our training methodology essentially permits aerial images to approach images from their corresponding ground-level scenes and natural language text within the CLIP space. Combining the trained aerial image encoder $s_\theta$ with the CLIP model allows us to achieve embeddings within a unified embedding space for goal contents that span diverse modalities (in particular, aerial images, ground-level images, and natural language text). We refer to this combined model as a *CLIP-based Multi-Modal Feature Extractor (CLIP-MMFE)*.

**RL-aligned representation learning using goal aware supervised pretraining of LLM.** Large Language Models (LLMs) are in general not proficient planners [39], but recent studies have effectively utilized the capabilities of LLMs to grasp abstract concepts of the world model dynamics in addressing decision-making challenges [35, 42, 1]. However, discrepancies between the knowledge of LLMs and the environment can lead to inaccuracies and constrain their functional effectiveness due to insufficient grounding.

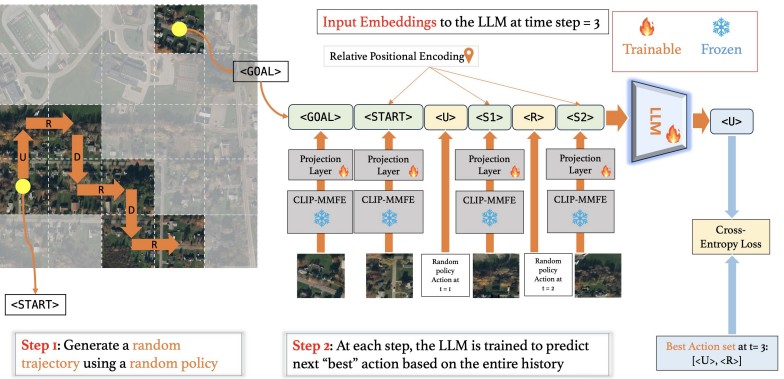

Figure 2: *GASP* strategy for pretraining LLMs for AGL.

In a GC-POMDP (cf. Section 3), we aim to learn a goal-conditioned latent representation that encompasses the complete history of observed state and action sequences, aiding the agent in decision-making within the partially observed environment. For this purpose, we leverage state-of-the-art LLMs, which excel in long-range autoregressive and sequence modeling tasks. However, employing such a model naively is not conducive to obtaining a latent representation that is advantageous for goal-conditioned active geo-localization (AGL). Therefore, we devise a *Goal-Aware Supervised Pretraining (GASP)* strategy that enables learning a history-aware goal-conditioned latent representation, which assists in decision-making for the subsequent policy. An overview of the GASP strategy is depicted in Figure 2, and involves a two-step training process. First, we generate a random sequence of movement actions. Each random sequence of length $t$ comprises all the previous observations and actions up to time $t$. A $t$-length random sequence is denoted $x_{h_t} = \{x_0, a_0, x_1, a_1, \ldots, x_t\}$. Second, we train the LLM on a sequence modeling task that involves predicting the optimal actions at time $t$ that will bring the agent closer to the goal location, based on the observed trajectory data $x_{h_t}$ and the goal content $x_g$. Accordingly, we train an LLM using the binary cross entropy loss defined as:

$$\mathcal{L}_{\text{BCE}} = \sum_{i=1}^{N} -(y_i \log(p_i) + (1 - y_i) \log(1 - p_i))$$
$$p_i = \sigma(\text{LLM}(o|x_{h_{i-1}}, g)), \text{ where } |p_i| = |\mathcal{A}| \tag{3}$$

Here $N$ is the length of the random sequence, $p_i$ is the predicted probability of actions at time step $i$ when $x_{h_{i-1}}$ and $g$ are given as the input to LLM, and $y_i$ encodes information regarding the set of actions $\mathcal{A}_i^{\text{opt}} \subset \mathcal{A}$ that are considered optimal at time step $i$, implying that these actions will lead the agent closer to the goal location. Each element of $y_i$ corresponds to an action in the set $\mathcal{A}$, so that $y_i = [y_i^{(1)}, y_i^{(2)}, \ldots, y_i^{(|\mathcal{A}|)}]$, where $y_i^{(j)} = 1$ if $j$'th action $\in \mathcal{A}_i^{\text{opt}}$, and otherwise $y_i^{(j)} = 0$. We perform extensive experiments to validate the efficacy of the GASP strategy (see Section 6), and refer to the appendix section K for details about the GASP architecture and training process.

**Planning.** So far we have focused on the learning of a history-aware, goal modality agnostic latent representation useful for the AGL task in partially observable environments. Now, we describe the approach for learning an effective policy that leverages the learned latent representation to address this GC-POMDP. We refer to the latent representation obtained from the LLM at time $t$ as $e_t^{\text{LLM}}(x_{h_t}, g)$. Formally, we aim to learn a policy $\pi$ that maximizes the expected discounted sum of rewards for any given goal $g \in \mathcal{G}$. To this end, we use an actor-critic style PPO algorithm [33] that involves learning both an *actor* (policy network, parameterized by $\zeta$) $\pi_\zeta : e_t^{\text{LLM}}(x_{h_t}, g) \to p(\mathcal{A})$ and a *critic* (value function, parameterized by $\eta$) $V_\eta : e_t^{\text{LLM}}(x_{h_t}, g) \to \mathbb{R}$ that approximates the true value $V^{\text{true}}(x_t, g) = \mathbb{E}_{a \sim \pi_\zeta(.|e_t^{\text{LLM}}(x_{h_t}, g))}[R(x_t, a, g) + \gamma V(\mathcal{T}(x_t, a), g)]$. We optimize both the actor and critic networks using the following loss function:

$$\mathcal{L}_t^{\text{planner}}(\zeta, \eta) = \mathbb{E}_t \left[ -\mathcal{L}^{\text{clip}}(\zeta) + \alpha \mathcal{L}^{\text{crit}}(\eta) - \beta \mathcal{H} \left[ \pi_\zeta(.|e_t^{\text{LLM}}(x_{h_t}, g)) \right] \right] \tag{4}$$

Here $\alpha$ and $\beta$ are hyperparameters, and $\mathcal{H}$ denotes entropy, so minimizing the final term of (4) encourages the actor to exhibit more exploratory behavior. The $\mathcal{L}^{\text{crit}}$ loss is used specifically to optimize the parameters of the critic network and is defined as a squared-error loss, i.e. $\mathcal{L}^{\text{crit}} = (V_\eta(e_t^{\text{LLM}}(x_{h_t}, g)) - V^{\text{true}}(x_t, g))^2$. The clipped surrogate objective $\mathcal{L}^{\text{clip}}$ is employed to optimize the parameters of the actor network while constraining the change to a small value $\epsilon$ relative to the old actor policy $\pi^{\text{old}}$ and is defined as:

$$\mathcal{L}^{\text{clip}}(\zeta) = \min \left\{ \frac{\pi_\zeta(.|e_t^{\text{LLM}}(x_{h_t}, g))}{\pi^{\text{old}}(.|e_t^{\text{LLM}}(x_{h_t}, g))} A^t, \text{clip} \left( 1 - \epsilon, 1 + \epsilon, \frac{\pi_\zeta(.|e_t^{\text{LLM}}(x_{h_t}, g))}{\pi^{\text{old}}(.|e_t^{\text{LLM}}(x_{h_t}, g))} )A^t \right) \right\}$$
$$A^t = r_t + \gamma r_{t+1} + \ldots + \gamma^{T-t+1} r_{T-1} - V_\eta(e_t^{\text{LLM}}(x_{h_t}, g)) \tag{5}$$

After every fixed update step, we copy the parameters of the current policy network $\pi_\zeta$ onto the old policy network $\pi^{\text{old}}$ to enhance training stability. All hyperparameter details for training the actor and critic network are in the appendix section K.

There are numerous options for crafting the reward function for the AGL task. One potential approach involves designing a sparse reward signal, where the agent only receives a positive reward upon reaching the goal location and receives either no or a negative reward otherwise. Nevertheless,

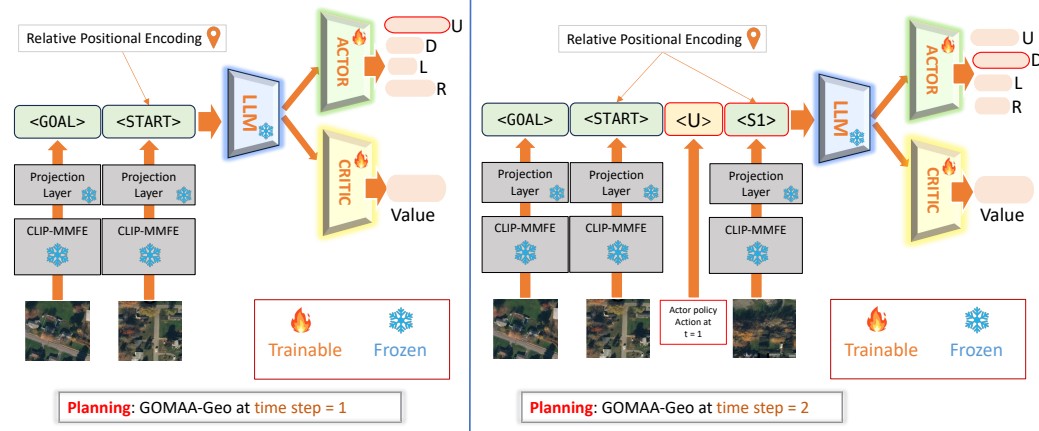

Figure 3: Our proposed *GOMAA-Geo* framework for goal modality agnostic active geo-localization.

incorporating a denser reward structure has been demonstrated to expedite learning and improve the efficacy of the learned policy (shown in section 6). Specifically, we formulate a dense reward function customized for the AGL task as follows:

$$R(x_{h_t}, a_t, x_g, x_{t+1}) = \begin{cases} 1, & \text{if } ||p_{t+1} - p_g||_2^2 < ||p_t - p_g||_2^2 \\ -1, & \text{if } ||p_{t+1} - p_g||_2^2 > ||p_t - p_g||_2^2 \ \lor \ x_{t+1} \in \{x_{h_t}\} \\ 2, & \text{if } x_{t+1} = x_g \end{cases} \quad (6)$$

In other words, our approach involves penalizing the agent when an action takes it away from the goal or when it revisits the same state. Conversely, the agent receives a positive reward when its current action brings it closer to the goal, with the highest reward granted when the action results in reaching the goal location.

**GOMAA-Geo.** Our full *GOMAA-Geo* framework integrates all the previously introduced components (see Figure 3). Initially, the aerial image encoder $s_\theta$ is trained to align the aerial image and CLIP embeddings. Next, the LLM is trained using the GASP strategy while maintaining $s_\theta$ and the CLIP model frozen. Finally, the LLM is also frozen, and only the actor and critic are trained using RL.

## 5 Experiments and Results

**Baselines.** In this and the subsequent section 6, we evaluate and analyze *GOMAA-Geo* and compare its performance against the following baseline approaches: **(i)** *Random policy* selects an action uniformly at random from the action set $\mathcal{A}$ at each time step; **(ii)** *AiRLoc* [24] is an RL-based model designed for uni-modal AGL tasks [24]. The approach involves training the policy using DRL and encoding the history of state observations using an LSTM [13]. Note that *AiRLoc* is *not* agnostic to the goal modality; **(iii)** *PPO policy* [33] selects actions based solely on the current observation; **(iv)** *Decision Transformer (DiT)* [8] is trained using a collection of offline optimal trajectories that span from randomly selected start to randomly selected goal grids.

**Evaluation metric.** We evaluate the proposed approaches based on the *success ratio* (SR), which is measured as the ratio of the number of successful localizations of the goal within a predefined exploration budget $\mathcal{B}$ to the total number of AGL tasks. We evaluate the SR of *GOMAA-Geo* and the baselines across different distances $\mathcal{C}$ from the start to the goal location. In the main paper, we analyze *GOMAA-Geo* with a $5 \times 5$ grid structure, $\mathcal{B} = 10$, and varying start-to-goal distance $\mathcal{C} \in \{4, 5, 6, 7, 8\}$. In the appendix section A, we conduct additional experiments across various grid configurations, each employing different values of $\mathcal{B}$ with varying $\mathcal{C}$.

**Datasets.** We primarily utilize the Massachusetts Buildings (Masa) dataset [20] for both the development and evaluation of *GOMAA-Geo* in settings where the goal content is provided as aerial imagery. The dataset is split into 70% for training and 15% each for validation and testing.

Many existing datasets containing paired aerial and ground-level images lack precise coordinate locations, which are pivotal for the AGL task. Furthermore, the ground-level images typically contain

Table 1: Evaluation with aerial image goals. *GOMAA-Geo* obtains the highest success ratio (SR).

| | Evaluation using Masa Dataset | | | | | Evaluation using MM-GAG Dataset | | | | |
|---|---|---|---|---|---|---|---|---|---|---|
| Method | $\mathcal{C}=4$ | $\mathcal{C}=5$ | $\mathcal{C}=6$ | $\mathcal{C}=7$ | $\mathcal{C}=8$ | $\mathcal{C}=4$ | $\mathcal{C}=5$ | $\mathcal{C}=6$ | $\mathcal{C}=7$ | $\mathcal{C}=8$ |
| Random | 0.1412 | 0.0584 | 0.0640 | 0.0247 | 0.0236 | 0.1412 | 0.0584 | 0.0640 | 0.0247 | 0.0236 |
| PPO | 0.1427 | 0.1775 | 0.1921 | 0.2269 | 0.2595 | 0.1489 | 0.1854 | 0.1879 | 0.2176 | 0.2432 |
| DiT | 0.2011 | 0.2956 | 0.3567 | 0.4216 | 0.4559 | 0.2023 | 0.2856 | 0.3516 | 0.4190 | 0.4423 |
| AiRLoc | 0.1786 | 0.1561 | 0.2134 | 0.2415 | 0.2393 | 0.1745 | 0.1689 | 0.2019 | 0.2156 | 0.2290 |
| **GOMAA-Geo** | **0.4090** | **0.5056** | **0.7168** | **0.8034** | **0.7854** | **0.4085** | **0.5064** | **0.6638** | **0.7362** | **0.7021** |

little to no meaningful information about the goal (e.g. images of roads, trees, and so on). In particular, to the best of our knowledge, no open-source dataset is currently available for evaluating the zero-shot generalizability of *GOMAA-Geo* across diverse goal modalities, such as ground-level images and natural language text. To alleviate this, we have collected a dataset from different regions across the world, which allows for specifying the goal content as aerial imagery, ground-level imagery, or natural language text. This dataset allows us to conduct proof-of-concept AGL experiments in contexts where the goal modality may vary at test time. Note that the data is only used for evaluation – training is done on Masa using only aerial imagery as a goal modality. We refer to this dataset as Multi-Modal Goal Dataset for Active Geolocalization (*MM-GAG*). It consists of 73 distinct search areas from different parts of the world. For each area, we select 5 pairs of start and goal locations corresponding to each start-to-goal distance $\mathcal{C}$ (resulting in 365 evaluation scenarios for each $\mathcal{C}$). We provide more details about the dataset in the appendix section L.

Finally, to further evaluate the zero-shot generalization capability, we also compare the *GOMAA-Geo* with baseline approaches using the xBD dataset introduced by Gupta et al. (2019). This dataset includes aerial images from different regions, both before (xBD-pre) and after (xBD-disaster) various natural disasters such as wildfires and floods. It is important to emphasize that all the results we present in this work – including the zero-shot generalization settings with the xBD-pre and xBD-post disaster datasets, as well as all across the three different goal modalities of the MM-GAG dataset – *are evaluated using a model trained exclusively on the Masa dataset* (where goals are always specified from an aerial perspective).

**Implementation details.** We provide comprehensive details on network architectures and training hyperparameters for each training stage in the appendix section K.

**Evaluation of *GOMAA-Geo*.** We initiate our evaluation of the proposed methods using the Masa and MM-GAG datasets with *aerial image* as goal modality. During the evaluation, for each AGL task, we randomly select 5 pairs of start and goal locations for each value of $\mathcal{C}$. We report the result in Table 1 and observe a significant performance improvement compared to the baseline methods, with success ratio (SR) improvements ranging from $129.00\%$ to $232.67\%$ relative to the baselines across various evaluation settings, showcasing the efficacy of *GOMAA-GEO* for active geo-localization (AGL) tasks where goals are provided in the form of aerial images.

We next evaluate the performance of *GOMAA-Geo* across *various goal modalities* using the MM-GAG dataset and present the results in Table 2. For this evaluation, we employ the model trained on the Masa dataset. We see that ***GOMAA-***

Table 2: *GOMAA-Geo* generalizes well across goal modalities.

| Goal Modality | $\mathcal{C}=4$ | $\mathcal{C}=5$ | $\mathcal{C}=6$ | $\mathcal{C}=7$ | $\mathcal{C}=8$ |
|---|---|---|---|---|---|
| Text | 0.4000 | 0.4978 | 0.6766 | 0.7702 | 0.6595 |
| Ground Image | 0.4383 | 0.5150 | 0.6808 | 0.7489 | 0.6893 |
| Aerial Image | 0.4085 | 0.5064 | 0.6638 | 0.7362 | 0.7021 |

***Geo* efficiently performs the AGL task across different goal modalities – with comparable performance observed across different modalities – despite only being trained with aerial views as goal modality.** This highlights the effectiveness of the CLIP-MMFE (cf. Section 4) module in learning modality-invariant representations. Further analyses of CLIP-MMFE are provided in Section 6. The experimental outcomes also demonstrate the zero-shot generalization capability of *GOMAA-Geo* across different goal modalities.

**Zero-shot generalization capabilities of *GOMAA-Geo*.** For additional assessments of *GOMAA-Geo*'s zero-shot generalizability, we employ trained *GOMAA-Geo* model exclusively trained on the Masa dataset and evaluate them on both non-disaster data from xBD-pre and disaster data from xBD-disaster. For fair evaluation, we ensure that the training data from Masa depicts geographical areas different from those in xBD. Moreover, in both pre-and post-disaster evaluation scenarios, the goal content is always presented to the agent as an aerial image captured *before* the disaster. For

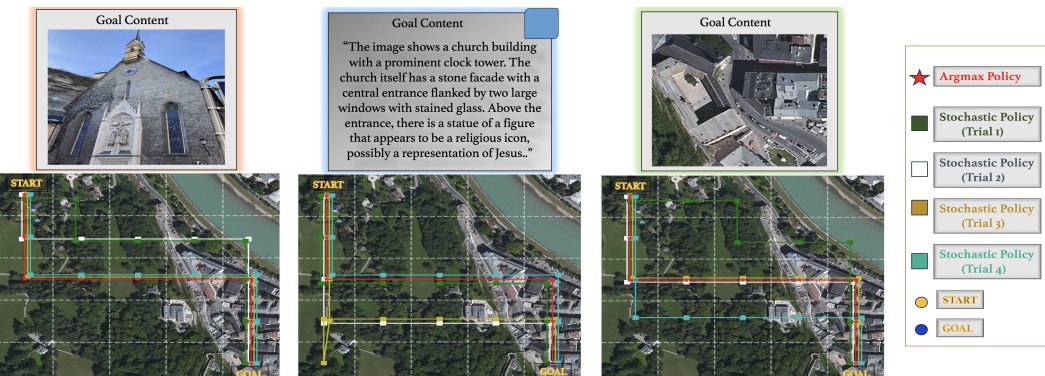

Figure 4: **Example exploration behavior of *GOMAA-Geo* across different goal modalities.** The stochastic policy selects actions probabilistically, whereas the *argmax* policy selects the action with the highest probability.

the xBD-disaster setup, this thus depicts the challenging scenario of localizing a goal whose visual description is provided prior to a disaster, which may look drastically different when exploring the scene after said disaster. The evaluation dataset comprises 800 distinct search areas from both xBD-pre and xBD-disaster. These 800 search areas are identical in both xBD-pre and xBD-disaster. For each of these areas, we randomly select 5 pairs of start and goal locations corresponding to each value of $\mathcal{C}$. We present the zero-shot generalization results using both xBD-pre and xBD-disaster in Table 3. The results show a substantial performance improvement *ranging between* $221.15\%$ *to* $346.83\%$ compared to the baseline approaches and justify the effectiveness of *GOMAA-Geo* in zero-shot generalization.

Table 3: *GOMAA-Geo* showcases superior zero-shot generalization than the alternatives in all settings.

| | Evaluation using xBD-pre Dataset | | | | | Evaluation using xBD-disaster Dataset | | | | |
|---|---|---|---|---|---|---|---|---|---|---|
| Method | $\mathcal{C}=4$ | $\mathcal{C}=5$ | $\mathcal{C}=6$ | $\mathcal{C}=7$ | $\mathcal{C}=8$ | $\mathcal{C}=4$ | $\mathcal{C}=5$ | $\mathcal{C}=6$ | $\mathcal{C}=7$ | $\mathcal{C}=8$ |
| Random | 0.1412 | 0.0584 | 0.0640 | 0.0247 | 0.0236 | 0.1412 | 0.0584 | 0.0640 | 0.0247 | 0.0236 |
| PPO | 0.1237 | 0.1262 | 0.1425 | 0.1737 | 0.2075 | 0.1132 | 0.1146 | 0.1292 | 0.1665 | 0.1953 |
| DiT | 0.1132 | 0.2341 | 0.3198 | 0.3664 | 0.3772 | 0.1012 | 0.2389 | 0.3067 | 0.3390 | 0.3543 |
| AiRLoc | 0.1191 | 0.1254 | 0.1436 | 0.1676 | 0.2021 | 0.1201 | 0.1298 | 0.1507 | 0.1631 | 0.1989 |
| **GOMAA-Geo** | **0.3825** | **0.4737** | **0.6808** | **0.7489** | **0.7125** | **0.4002** | **0.4632** | **0.6553** | **0.7391** | **0.6942** |

## 6 Further Analyses and Ablation Studies

**Effectiveness of the CLIP-MMFE module.** In addition to the quantitative results in Table 2, we here conduct a qualitative analysis to assess the effectiveness of the CLIP-MMFE module in learning a modality-agnostic goal representation. For this purpose, we employ the *GOMAA-Geo* model trained on the Masa dataset. During the evaluation, we maintain a fixed search area with identical start and goal locations while varying the goal modality, and then compare the exploration behavior of *GOMAA-Geo* in each scenario. Our observations (Figure 4) reveal that exploration behaviors are consistent across different goal modalities. This suggests that the learned representation of the goal token remains consistent regardless of the goal modality, given that the other components of the *GOMAA-Geo* framework remain fixed in each scenario. Additional visualizations are in the section B.

**Importance of learning a goal conditioned policy.** To investigate the significance of goal information in the *GOMAA-Geo* framework, we assess a *GOMAA-Geo* variant – denoted *Mask-GOMAA* – where we mask out the

Table 4: Providing goal information is crucial.

| Method | $\mathcal{C}=4$ | $\mathcal{C}=5$ | $\mathcal{C}=6$ | $\mathcal{C}=7$ | $\mathcal{C}=8$ |
|---|---|---|---|---|---|
| Mask-GOMAA | 0.2913 | 0.3566 | 0.4912 | 0.5200 | 0.5478 |
| **GOMAA-Geo** | **0.4090** | **0.5056** | **0.7168** | **0.8034** | **0.7854** |

goal token and compare its performance against the full *GOMAA-Geo*. Results on the Masa dataset are presented in Table 4. We observe a substantial drop in performance ranging from $40.40\%$ to $54.50\%$ across different evaluation settings, underscoring the critical role of goal information in learning an effective policy for AGL.

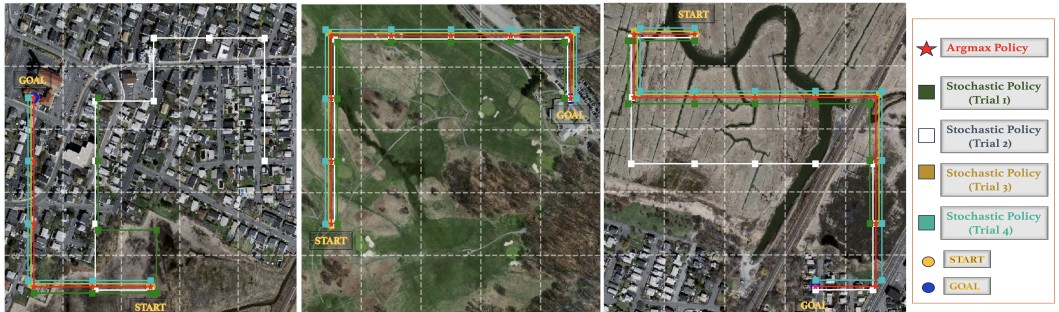

Figure 5: Examples of successful exploration behaviors of *GOMAA-Geo*.

**Importance of the planner module.** To evaluate the significance of the planner module, we conduct experiments when removing it from *GOMAA-Geo* and compare the performance of this modified version, termed *LLM-Geo*, with the original *GOMAA-Geo*. The only distinction between *LLM-Geo* and *GOMAA-Geo* is the presence of the planner module in the latter.

We compare their performances across various evaluation settings using the Masa dataset, with results presented in Table 5. We observe that the performance of *LLM-Geo* is significantly inferior to *GOMAA-Geo*, with performance gaps ranging from $75.46\%$ to

Table 5: On the importance of the planner module.

| Method | $\mathcal{C} = 4$ | $\mathcal{C} = 5$ | $\mathcal{C} = 6$ | $\mathcal{C} = 7$ | $\mathcal{C} = 8$ |
|---|---|---|---|---|---|
| LLM-Geo | 0.2331 | 0.2591 | 0.3121 | 0.3967 | 0.4051 |
| **GOMAA-Geo** | **0.4090** | **0.5056** | **0.7168** | **0.8034** | **0.7854** |

$129.66\%$ across various evaluation settings. The empirical results indicate that relying solely on the LLM is insufficient for solving tasks that involve planning, highlighting the importance of combining an LLM – which excels at capturing history – with a planning module that learns to make decisions while considering future outcomes.

**Visualizing the exploration behavior of GOMAA-Geo.** In Figure 5, we present a series of exploration trajectories generated using the trained stochastic policy for a specific start and goal pair. Alongside these trajectories, we include an additional exploration trajectory obtained using the deterministic (argmax) policy. Please refer to the appendix for several additional qualitative results.

**Efficacy of GASP pretraining.** We assess the effectiveness of GASP (cf. Section 4) by comparing the performance of the original *GOMAA-Geo* with a version of *GOMAA-Geo* that involves pre-training an LLM using a commonly used input token masking-based autoregressive modeling task tailored for AGL. We call the resulting *GOMAA-Geo* model *R*elative *P*osition to *G*oal aware *GOMAA* (*RPG-GOMAA*). Details of the *RPG-GOMAA* framework, along with the masking-based LLM pre-training strategy, are provided in the appendix section F. We compare the performance of *GOMAA-Geo* with *RPG-GOMAA* on the Masa dataset and present the results in Table 6. Our findings indicate that RPG-GOMAA *shows a significant decline in performance compared to* GOMAA-Geo *as the distance between the start to the goal* $\mathcal{C}$ *increases*, which highlights the efficacy of GASP in learning an effective

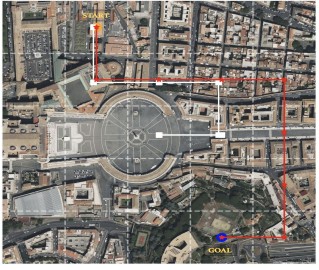

Figure 6: *GOMAA-Geo* (red) vs. GOMAA-Geo w/o history (white).

history-aware policy for AGL. We also compare the zero-shot generalization capabilities of these methods using the xBD-disaster dataset and report the results in the appendix section E.

Furthermore, we qualitatively assess GASP's effectiveness in learning a history-aware representation for planning. To achieve this, we choose a task where *GOMAA-Geo* successfully locates the goal. We then remove the context by masking out all

Table 6: GASP (bottom) yields the best results overall.

| Method | $\mathcal{C} = 4$ | $\mathcal{C} = 5$ | $\mathcal{C} = 6$ | $\mathcal{C} = 7$ | $\mathcal{C} = 8$ |
|---|---|---|---|---|---|
| RPG-GOMAA | **0.4116** | **0.5167** | 0.6589 | 0.7643 | 0.7023 |
| **GOMAA-Geo** | 0.4090 | 0.5056 | **0.7168** | **0.8034** | **0.7854** |

previously visited states and actions except for the current state $x_t$ and goal state $x_g$, and observe the action selected by the policy. We compare this action to what the policy selects when the entire history is provided as input to the LLM (i.e., the default *GOMAA-Geo*). From Figure 6 we see that the policy takes optimal actions with context but suboptimal actions without it, which suggests that

the LLM trained via GASP effectively learns a history-aware representation suitable for planning. Additional visualizations are in the appendix section E.

**Failure cases of *GOMAA-Geo*.** Some failure cases are shown in Figure 7. In scenarios where the goal patch is very similar to many other patches in the search space, *GOMAA-Geo* can become unsuccessful (such scenarios are very confusing even for humans).

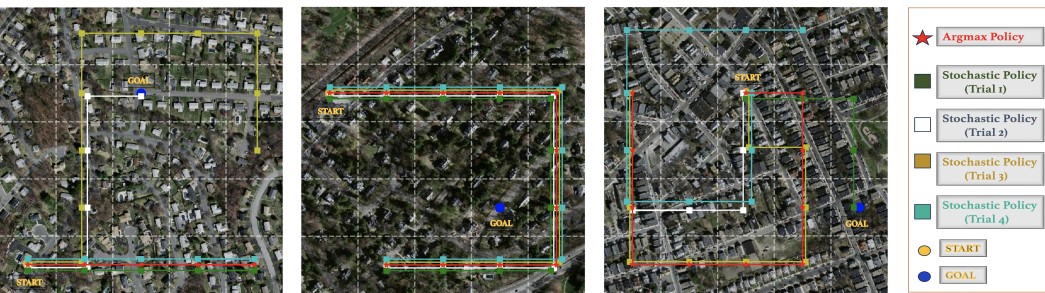

Figure 7: Examples of failure cases of *GOMAA-Geo*. Common for each example is that the goal patch is very similar to many of the surrounding patches, which makes the search problems particularly challenging.

**Effectiveness of proposed dense reward function.** Here we investigate the effectiveness of the proposed reward function (see Equation 6 in the main paper) for learning an efficient policy for active geo-localization tasks. We achieve this by training the planner module with a sparse reward function that assigns a positive reward of 1 exclusively upon reaching the desired goal location, and 0 otherwise. Throughout this analysis, we maintain all other components of the original *GOMAA-Geo* framework and training hyperparameters unchanged. The resulting framework is referred to as *Sparse-GOMAA*. We proceed to compare the performance of *GOMAA-Geo* and *Sparse-GOMAA*

Table 7: On the effects of using a dense reward. Left: Evaluations on Masa. Right: Zero-shot evaluation using xBD-disaster. Using the proposed dense reward (bottom row) yields significantly better results than if training based on a sparse reward (top row).

| | Test with $\mathcal{B} = 10$ in $5 \times 5$ setting using Masa | | | | | Test with $\mathcal{B} = 10$ in $5 \times 5$ setting using xBD-disaster | | | | |
|---|---|---|---|---|---|---|---|---|---|---|
| Method | $\mathcal{C} = 4$ | $\mathcal{C} = 5$ | $\mathcal{C} = 6$ | $\mathcal{C} = 7$ | $\mathcal{C} = 8$ | $\mathcal{C} = 4$ | $\mathcal{C} = 5$ | $\mathcal{C} = 6$ | $\mathcal{C} = 7$ | $\mathcal{C} = 8$ |
| Sparse-GOMAA | 0.3562 | 0.4312 | 0.6009 | 0.7318 | 0.6978 | 0.3615 | 0.3842 | 0.5367 | 0.6455 | 0.6290 |
| **GOMAA-Geo** | **0.4090** | **0.5056** | **0.7168** | **0.8034** | **0.7854** | **0.4002** | **0.4632** | **0.6553** | **0.7391** | **0.6942** |

under various evaluation settings using the Masa dataset. The findings of this evaluation are presented in Table 7 (left). We observe *a significant and consistent drop in performance in terms of SR across all evaluation settings, ranging from 10.71% to 22.10%*. The observed outcomes of the experiment suggest that dense reward is indeed helpful for learning an efficient policy for active geo-localization. Additionally, we assess the zero-shot generalizability of *Sparse-GOMAA* using xBD-disaster data and compare its performance against *GOMAA-Geo* (right part of Table 7). We notice that the performance gap widens further, which indicates that the policy learned with the sparse reward function exhibits limited generalization capability.

# 7 Conclusions

We have introduced *GOMAA-Geo*, a goal modality agnostic active geo-localization agent designed for zero-shot generalization across different goal modalities. Our method integrates cross-modality contrastive learning to align representations across modalities, supervised foundation model pretraining, and reinforcement learning to develop highly effective navigation and localization policies. Extensive evaluations demonstrate that *GOMAA-Geo* outperforms other approaches and generalizes across datasets – such as disaster-hit areas without prior exposure during training – and goal modalities (ground-level imagery, as well as textual descriptions), despite being trained solely with aerial view goals. We hope our framework will find applications ranging from search-and-rescue to environmental monitoring.

## Acknowledgments

This research was partially supported by the NSF (IIS-1905558, IIS-2214141, CNS-2310470, CCF-2403758), Amazon, NVIDIA, and the Taylor Geospatial Institute.

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

**GOMAA-Geo: GOal Modality Agnostic Active Geo-localization (Appendix)**

In this supplementary material, we provide additional insights into our *GOMAA-Geo* framework. Additional visualizations of the exploration behavior of *GOMAA-Geo* across 3 different goal modalities are presented in Section B. We also evaluate *GOMAA-Geo* across different grid sizes and report the results in Section A. We then analyze the trade-off in learning modality-specific vs. modality-invariant goal representation for active geo-localization in Section C. In Section D, we provide more visualizations of the exploration behavior of *GOMAA-Geo*. More qualitative evaluations and zero-shot generalizability of the GASP pre-training strategy are discussed in Section E. In Section I, we compare the search performance for different choices of LLM architecture. Furthermore, in Section H, we study the importance of sampling strategy for selecting start-to-goal distance in policy training. We also evaluate and compare the performance of *GOMAA-Geo* with varying search budget $\mathcal{B}$ while keeping the value of $\mathcal{C}$ fixed and present the results in Section G. In Section J we present visualizations of the exploration behavior of *GOMAA-Geo* in disaster-hit regions, even though *GOMAA-Geo* has been trained exclusively on pre-disaster data (thus these visualizations illustrate examples of *GOMAA-Geo*'s zero-shot generalization performance). In Section K, we provide all the implementation details, including training hyperparameters, network architecture, and computing resources used to train *GOMAA-Geo*. Details of our curated dataset MM-GAG is discussed in Section L. In Section M we present the active geo-localization performance of *GOMAA-Geo* along with the baseline approaches across multiple trials using a boxplot. We provide the details of the *RPG-GOMAA* framework in Section F. Finally, we include a brief discussion on the limitations and broader impacts of *GOMAA-Geo*.

## A    Evaluation of GOMAA-Geo across Different Grid Sizes

Here we evaluate the performance of *GOMAA-Geo* in a $10 \times 10$ grid setting using the Masa dataset and report the result in Table 8. We also assess the zero-shot generalizability of *GOMAA-Geo* in a $10 \times 10$ grid settings using the xBD-disaster dataset, and depict the findings in Table 9. Note that in all these evaluations on larger grid sizes, we use the *GOMAA-Geo* model which was trained exclusively on the Masa dataset with a $5 \times 5$ grid configuration. Consequently, *all results reflect the model's ability to generalize from smaller to larger grid sizes.*

Table 8: Evaluations on another grid size, given models trained exclusively on smaller grid sizes. *GOMAA-Geo* significantly outperforms the compared methods.

| | Test with $\mathcal{B} = 20$ in $10 \times 10$ setting | | | | |
|---|---|---|---|---|---|
| Method | $\mathcal{C} = 12$ | $\mathcal{C} = 13$ | $\mathcal{C} = 14$ | $\mathcal{C} = 15$ | $\mathcal{C} = 16$ |
| Random | 0.0314 | 0.0280 | 0.0157 | 0.0112 | 0.0101 |
| DiT | 0.0923 | 0.1015 | 0.0876 | 0.0864 | 0.0932 |
| **GOMAA-Geo** | **0.2427** | **0.2360** | **0.2438** | **0.2685** | **0.2685** |

The experimental outcomes suggest that similar to the $5 \times 5$ settings, we observe a significant improvement in performance compared to the baselines ranging from approximately $133.74\%$ to $188.80\%$ across various evaluation settings (including zero-shot generalization performance) in the $10 \times 10$ grid setting. Furthermore, results are as expected lower overall in the larger grid setting, which illustrates the difficult nature and motivates further research into our proposed active geo-localization setup.

Table 9: Zero-shot evaluation on another grid size, using xBD-disaster, given models trained exclusively on smaller grid sizes and on another dataset (Masa). *GOMAA-Geo* significantly outperforms the compared methods here as well.

| | Test with $\mathcal{B} = 20$ in $10 \times 10$ setting | | | | |
|---|---|---|---|---|---|
| Method | $\mathcal{C} = 12$ | $\mathcal{C} = 13$ | $\mathcal{C} = 14$ | $\mathcal{C} = 15$ | $\mathcal{C} = 16$ |
| Random | 0.0314 | 0.0280 | 0.0157 | 0.0112 | 0.0101 |
| DiT | 0.0912 | 0.0767 | 0.1010 | 0.1243 | 0.1165 |
| **GOMAA-Geo** | **0.2475** | **0.2162** | **0.2500** | **0.2650** | **0.2487** |

## B    More Visualizations of Exploration Behavior of *GOMAA-Geo* across different Goal Modalities

In this section, we conduct further qualitative analyses in order to evaluate the effectiveness of the CLIP-MMFE module (cf. Section 4) in learning modality-agnostic goal representations. Similar to Figure 4 (main paper), here we use the *GOMAA-Geo* model trained on the Masa dataset – thus the agent is trained only in settings where goals are specified as aerial views. During evaluation, we keep the start and goal locations fixed while varying the goal modality, and then compare the exploration behavior of *GOMAA-Geo* in each scenario. These qualitative results (shown in Figure 8-9) reveal that exploration behaviors are consistent across different goal modalities. This indicates that the learned representation of the goal token remains consistent regardless of the goal modality, given that the other components of the *GOMAA-Geo* framework remain fixed in each scenario.

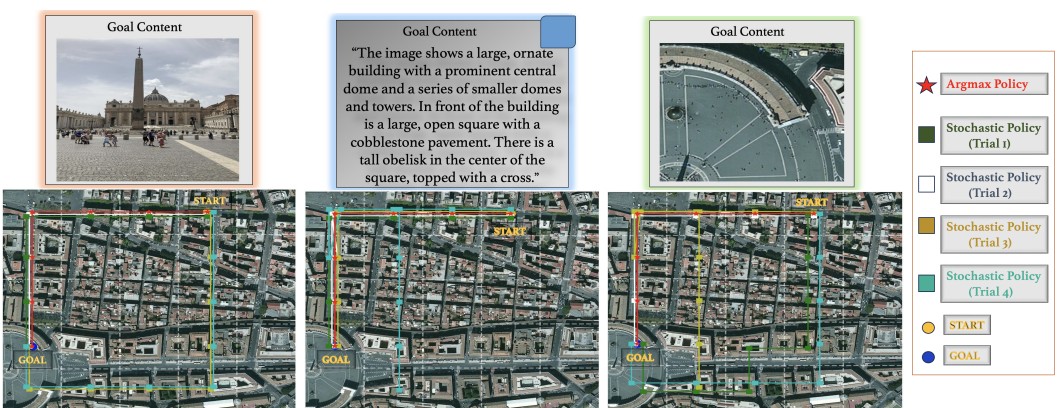

Figure 8: Example exploration behavior of *GOMAA-Geo* across different goal modalities. In this case, the agent successfully reaches the goal for all three goal modalities in the minimum number of steps (red line).

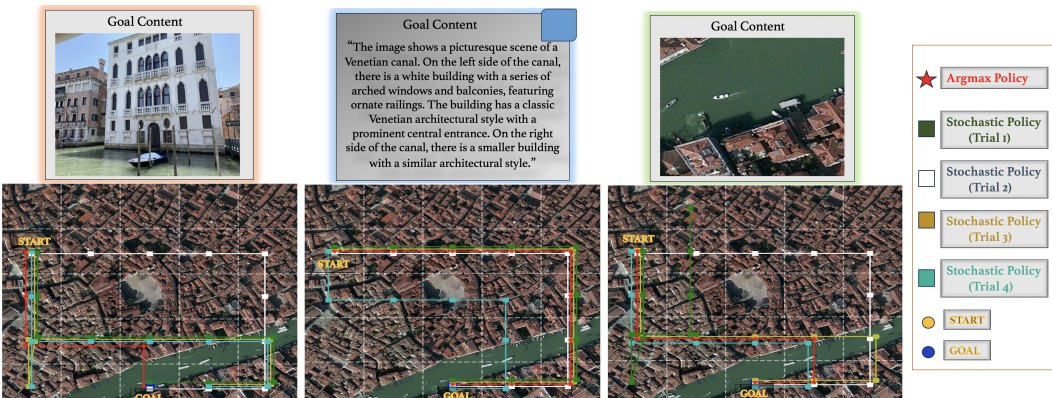

Figure 9: Example exploration behavior of *GOMAA-Geo* across different goal modalities. In this case, the agent successfully reaches the goal for all three goal modalities in the minimum number of steps (red line).

## C    Trade-off: Modality-specific vs. Modality-invariant Goal Representation in Active Geolocalization

Learning a modality-agnostic goal representation allows us to address the AGL problem across diverse goal modalities. Specifically, we achieve a similar success ratio across all 3 diverse goal modalities as reported in Table 2 (main paper). However, aligning representations across different modalities may not fully preserve the representational capabilities of the original modality.

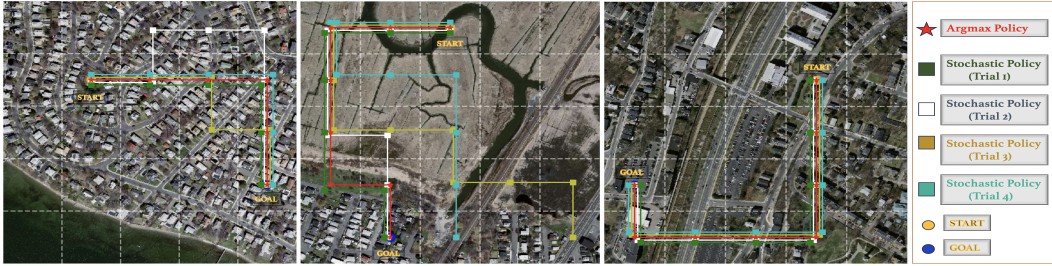

Figure 10: Example exploration behaviors of *GOMAA-Geo*. The agent is successful in all cases.

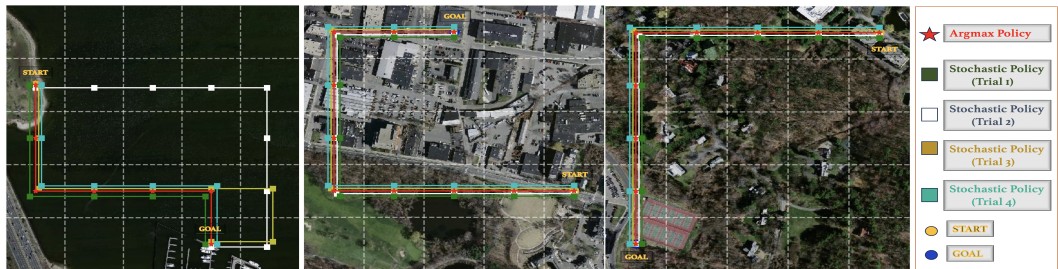

Figure 11: Example exploration behaviors of *GOMAA-Geo*. The agent is successful in all cases.

To investigate this hypothesis, we replace the CLIP-MMFE module of the *GOMAA-Geo* framework with *SatMAE* [9] (while keeping all other modules of the *GOMAA-Geo* framework unchanged), which is specifically designed to learn useful latent representations for downstream tasks involving satellite images. This modified framework is referred to as *SatMAE-Geo*.

Table 10: **Modality-specific (top) vs. modality-invariant agents (bottom)**. Despite our *GOMAA-Geo* being modality-agnostic, it still achieves comparable results to the modality-specific approach *SatMAE-Geo* in the setting which *SatMAE-Geo* is specifically trained on.

| | Test with $\mathcal{B} = 10$ in $5 \times 5$ setting | | | | |
|---|---|---|---|---|---|
| Method | $\mathcal{C} = 4$ | $\mathcal{C} = 5$ | $\mathcal{C} = 6$ | $\mathcal{C} = 7$ | $\mathcal{C} = 8$ |
| SatMAE-Geo | **0.4329** | **0.5221** | 0.7016 | 0.7914 | **0.7992** |
| **GOMAA-Geo** | 0.4090 | 0.5056 | **0.7168** | **0.8034** | 0.7854 |

We then compare the performance of our modality-invariant *GOMAA-Geo* with the modality-specific *SatMAE-Geo* using the Masa dataset, where the goal content is always provided as an aerial image. The results are presented in Table 10. Interestingly, we observe that the performance of *GOMAA-Geo* is only slightly inferior compared to *SatMAE-Geo* (in some evaluation settings, e.g. for $\mathcal{C} = 5$), despite *SatMAE-Geo* being targeted specifically towards AGL tasks in which goals are specified as aerial images. These findings suggest that on the one hand, learning modality-agnostic representations are beneficial for addressing active geo-localization problems across diverse goal modalities, on the other hand, they are equally competitive with models that are designed to solve modality-specific active geo-localization tasks, such as SatMAE-Geo.

## D  More Visualizations of Exploration Behavior of *GOMAA-Geo*

In this section,we provide additional visualizations of the exploration behavior of *GOMAA-Geo* across various geospatial regions. These visualizations are obtained using the Masa dataset with the goal specified as aerial images. Specifically, we present a series of exploration trajectories generated using the trained stochastic policy for a specific start and goal pair. Alongside these trajectories, we include an additional exploration trajectory obtained using the deterministic (argmax) policy. We depict the visualizations in Figure 10, 11, and 12. These visualizations allow us to compare and contrast the behaviors resulting from these different action selection strategies.

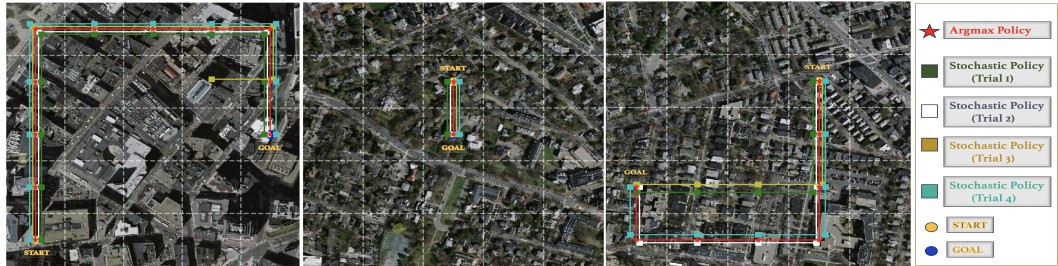

Figure 12: Example exploration behavior of *GOMAA-Geo*. The agent is successful in all cases.

## E More Qualitative Evaluation and Zero-Shot generalizability of *GASP*

We further analyze the effectiveness of *GASP* strategy in zero-shot generalization setting. Similar to table 6 in the main paper, here we compare the performance of *GOMAA-Geo* with *RPG-GOMAA* using the xBD-disaster dataset, while both these competitive approaches are being trained on the Masa dataset. Note that, during evaluation, the goal is presented as pre-disaster top-view imagery. We report the result in Table 11. We observe

Table 11: More on the efficacy of GASP. Zero-shot evaluation using xBD-disaster.

| | Test with $\mathcal{B} = 10$ in $5 \times 5$ setting | | | | |
|---|---|---|---|---|---|
| Method | $\mathcal{C} = 4$ | $\mathcal{C} = 5$ | $\mathcal{C} = 6$ | $\mathcal{C} = 7$ | $\mathcal{C} = 8$ |
| RPG-GOMAA | **0.4276** | 0.4519 | 0.6003 | 0.6754 | 0.6398 |
| GOMAA-Geo | 0.4002 | **0.4632** | **0.6553** | **0.7391** | **0.6942** |

that *the* RPG-GOMAA *method exhibits a significant drop in performance across different evaluation setups compared to* GOMAA-Geo *(specifically, we observe a larger performance gap for higher values of $\mathcal{C}$).* The findings presented in table 11 showcase the importance of GASP in learning an efficient zero-shot generalizable policy for AGL.

Similar to Figure 6 in the main paper, in this section, we present a further qualitative evaluation of the efficacy of the *GASP* strategy for planning. We compare the exploration behavior of *GOMAA-Geo*, which employs the *GASP* strategy during LLM pre-training, with the exploration behavior of *GOMAA-Geo* when the context is removed by masking out all previously visited states and actions, leaving only the current state $x_t$ and goal state $x_g$. We visualize and compare these exploration behaviors with several examples, as depicted in Figure 13.

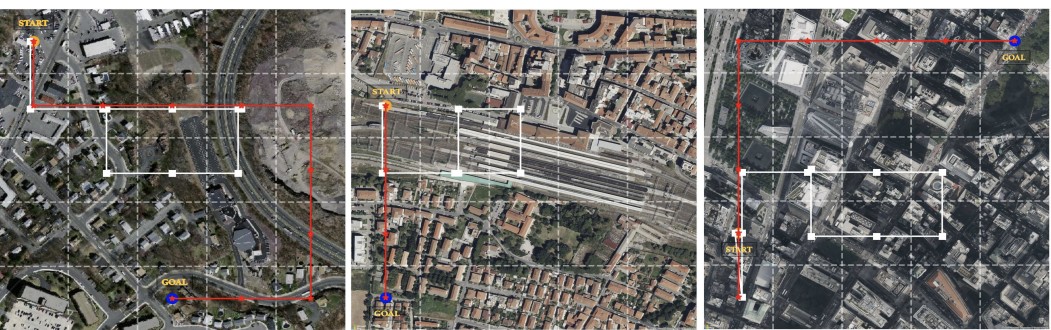

Figure 13: Visualization of the efficacy of GASP strategy for planning. *GOMAA-Geo* (red) outperforms *GOMAA-Geo* without history (white).

## F Details of the RPG-GOMAA Framework

The RPG-GOMAA framework is similar to GOMAA-Geo but differs in how the LLM is trained. In GOMAA-Geo, the LLM is trained using GASP. In contrast, the RPG-GOMAA framework employs a different training approach for the LLM. Initially, we generate an optimal sequence of states and actions using a random policy. Then, we randomly mask state and action tokens within this sequence. The LLM is trained to predict gradient vectors at each masked state token and the correct action at

each masked action token in an autoregressive manner. The gradient is calculated as the ratio of vertical to horizontal displacement. We compute all possible gradient vectors for a 5x5 grid, resulting in 47 unique gradient configurations. The pretraining strategy for masked state tokens involves predicting the correct gradient category out of these 47 categories, given the history and goal token.

## G  Performance Comparison of *GOMAA-Geo* with Varying Search Budget $\mathcal{B}$

In this section, we analyze the performance of GOMAA-Geo for fixed values of $\mathcal{C}$ while varying the search budget $\mathcal{B}$. Specifically, we conduct experiments with $\mathcal{C} = 5$ and $\mathcal{C} = 6$. For each setting, we vary $\mathcal{B}$ as follows: $\mathcal{B} \in \{\mathcal{C}, \mathcal{C} + 2, \mathcal{C} + 4, \mathcal{C} + 6, \mathcal{C} + 8\}$. We perform 5 independent experimental trials for each configuration and present the results using error bars in the boxplot shown in Figure 14. We observe that GOMAA-Geo achieves a higher success ratio as search budget $\mathcal{B}$ increases. We see the exactly same trend for different values of $\mathcal{C}$.

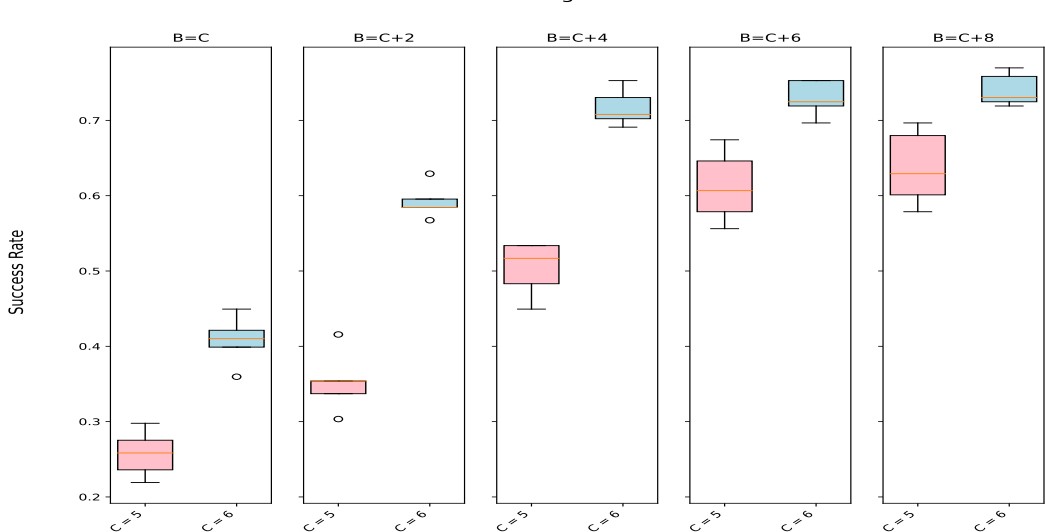

Figure 14: Performance of *GOMAA-Geo* across different values of search budget $\mathcal{B}$.

## H  Importance of Sampling Strategy for Selecting Start and Goal in Policy Training

During the training phase of the planner module, we construct training tasks by uniformly sampling distances between the start and goal locations. We then investigate how this sampling strategy affects policy training. To analyze this, we train the planner module using a curated set of training tasks where distances from start to goal are randomly sampled. We evaluate the performance of both

Table 12: On the importance of sampling. Left: Test using Masa. Right: Zero-shot evaluation using xBD-disaster.

| Sampling | Test with $\mathcal{B} = 10$ in $5 \times 5$ setting using Masa | | | | | Test with $\mathcal{B} = 10$ in $5 \times 5$ setting using xBD-disaster | | | | |
| --- | --- | --- | --- | --- | --- | --- | --- | --- | --- | --- |
| | $\mathcal{C} = 4$ | $\mathcal{C} = 5$ | $\mathcal{C} = 6$ | $\mathcal{C} = 7$ | $\mathcal{C} = 8$ | $\mathcal{C} = 4$ | $\mathcal{C} = 5$ | $\mathcal{C} = 6$ | $\mathcal{C} = 7$ | $\mathcal{C} = 8$ |
| Random | **0.4913** | **0.5567** | 0.6858 | 0.7412 | 0.6908 | **0.4758** | **0.4832** | 0.6054 | 0.6532 | 0.6178 |
| **Uniform** | 0.4090 | 0.5056 | **0.7168** | **0.8034** | **0.7854** | 0.4002 | 0.4632 | **0.6553** | **0.7391** | **0.6942** |

policies using the Masa dataset and compare their zero-shot generalization capabilities using the xBD-disaster dataset, presenting the results in Table 12. Our observations reveal that the policy trained with the uniform sampling strategy performs better on average across all evaluation settings (including the zero-shot generalization setting) compared to the policy trained with random sampling.

This outcome is intuitive because random sampling tends to bias the training toward certain distance ranges from start to goal. For instance, in a 5x5 grid setting, there are more samples with distances of 4 or 5 compared to samples with distances of 7 or 8. Consequently, the policy trained with random sampling is biased toward distances of 4 to 5 and exhibits poorer performance for distances of 7 to 8. To alleviate this issue, during the construction of training tasks, we adopt a uniform sampling strategy when selecting the distances between the start and goal for policy training.

## I    Performance Comparison for Different Choices of LLM Architecture

In this section, we assess the performance of *GOMAA-Geo* for different choices of LLM architectures: *Gemma* [37], *Falcon* [23], *GPT-2* [7], *Mamba* [27], *Llama-2* [29], and *Mistral* [14]. Note that we specifically choose decoder-only causal language models for this comparison, aligning with the requirements of the *GOMAA-Geo* framework. We conduct this assessment using the Masa dataset. Additionally, we examine the zero-shot generalizability of different models using the xBD-disaster dataset and also compare performance across different goal modalities using the MM-GAG dataset. We present the findings in tables 14, 15, 13. Our experimental outcome reveals that *Gemma* outperforms other models when evaluated using the Masa dataset. On the other hand, *Falcon* outperforms other models when evaluated using the xBD-disaster dataset in the zero-shot generalization setting, as reported in table 14. Conversely, the performance of *GOMAA-Geo* is notably lower when employing *Mistral* or *Llama-2* as the LLM model, showing the least favorable outcomes across all evaluation settings. We have identified exploring why representation learned via certain LLM models is more effective for planning compared to others as an important area for future research. We also observe that the performance of *GOMAA-Geo* is superior across different goal modalities when utilizing Gemma, GPT-2, Mamba, or Falcon as the LLM model as shown in table 15 and 13. Given its performance across various evaluation settings, we choose Falcon as the default choice for *GOMAA-Geo* framework due to its superior generalization capability compared to other LLM models.

Table 13: Results using different LLM models. Here, natural language text is used as goal modality.

| Test with $\mathcal{B} = 10$ in $5 \times 5$ setting using MM-GAG | | | | |
|---|---|---|---|---|
| Method | $\mathcal{C} = 5$ | $\mathcal{C} = 6$ | $\mathcal{C} = 7$ | $\mathcal{C} = 8$ |
| Gemma | **0.5319** | 0.5659 | 0.6851 | 0.5702 |
| GPT-2 | 0.4936 | 0.6170 | 0.7617 | 0.7532 |
| Mamba | 0.4851 | 0.6085 | **0.8170** | **0.7574** |
| Falcon | 0.4978 | **0.6766** | 0.7702 | 0.6595 |
| Llama-2 | 0.3659 | 0.4212 | 0.4425 | 0.3617 |
| Mistral | 0.3276 | 0.3829 | 0.4085 | 0.3829 |

Table 14: Evaluations using different LLM models. Left: Evaluations on Masa. Right: Zero-shot evaluation using xBD-disaster.

| | Test with $\mathcal{B} = 10$ in $5 \times 5$ setting | | | | Test with $\mathcal{B} = 10$ in $5 \times 5$ setting | | | |
|---|---|---|---|---|---|---|---|---|
| LLM | $\mathcal{C} = 5$ | $\mathcal{C} = 6$ | $\mathcal{C} = 7$ | $\mathcal{C} = 8$ | $\mathcal{C} = 5$ | $\mathcal{C} = 6$ | $\mathcal{C} = 7$ | $\mathcal{C} = 8$ |
| Gemma | 0.5044 | **0.7191** | **0.8966** | **0.8539** | **0.5075** | 0.6400 | 0.7462 | 0.6287 |
| GPT-2 | **0.5191** | 0.6842 | 0.8247 | 0.8011 | 0.4650 | 0.6450 | 0.7262 | 0.6812 |
| Mamba | 0.4415 | 0.5258 | 0.6326 | 0.7663 | 0.4011 | 0.5393 | 0.6932 | 0.6842 |
| Falcon | 0.5056 | 0.7168 | 0.8034 | 0.7854 | 0.4737 | **0.6850** | **0.7800** | **0.7125** |
| Llama-2 | 0.4393 | 0.5809 | 0.6011 | 0.5460 | 0.3612 | 0.4562 | 0.4013 | 0.3662 |
| Mistral | 0.4168 | 0.4752 | 0.5337 | 0.4853 | 0.3625 | 0.4162 | 0.4287 | 0.4075 |

## J    Visualizations of Exploration Behavior of *GOMAA-Geo* in Disaster-Hit Regions (Zero-shot Generalization Setting)

In this section, we visualize the exploration behavior of *GOMAA-Geo* in a zero-shot generalization setting. We train *GOMAA-Geo* using pre-disaster data and then visualize its exploration behavior with post-disaster data while providing the goal as a pre-disaster aerial image. These visualizations, shown in Figure 15, highlight the effectiveness of *GOMAA-Geo* in addressing AGL tasks in zero-shot generalization scenarios.

Table 15: Evaluations using different LLM models on the MM-GAG Dataset. Left: Goals specified as ground-level images. Right: Goals specified as aerial images.

| | Test with $\mathcal{B} = 10$ in $5 \times 5$ setting | | | | Test with $\mathcal{B} = 10$ in $5 \times 5$ setting | | | |
| LLM | $\mathcal{C} = 5$ | $\mathcal{C} = 6$ | $\mathcal{C} = 7$ | $\mathcal{C} = 8$ | $\mathcal{C} = 5$ | $\mathcal{C} = 6$ | $\mathcal{C} = 7$ | $\mathcal{C} = 8$ |
|---|---|---|---|---|---|---|---|---|
| Gemma | **0.5702** | 0.6510 | 0.7702 | 0.7021 | **0.5519** | 0.6612 | 0.7545 | 0.6890 |
| GPT-2 | 0.5148 | 0.5702 | 0.7957 | **0.7659** | 0.5213 | 0.5543 | 0.7687 | **0.7467** |
| Mamba | 0.4619 | 0.5932 | **0.8222** | 0.7651 | 0.4554 | 0.5832 | **0.8189** | 0.7323 |
| Falcon | 0.5150 | **0.6808** | 0.7489 | 0.6893 | 0.5064 | **0.6638** | 0.7362 | 0.7021 |
| Llama-2 | 0.3489 | 0.3744 | 0.3915 | 0.3319 | 0.3287 | 0.3719 | 0.4053 | 0.3542 |
| Mistral | 0.3617 | 0.3829 | 0.4085 | 0.3829 | 0.3745 | 0.3912 | 0.4221 | 0.3980 |

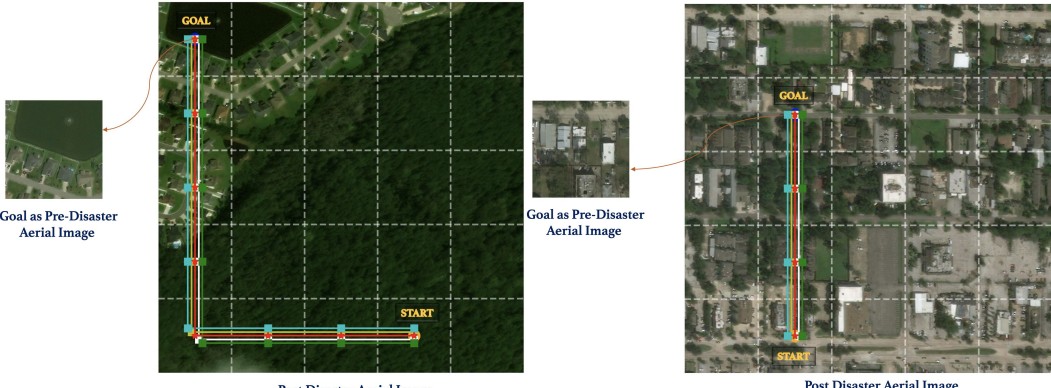

Figure 15: Exploration behavior of *GOMAA-Geo* in zero-shot generalization setting. The agent is successful in all cases.

## K   Implementation Details

In this section, we detail the training process for *GOMAA-Geo*. The proposed *GOMAA-Geo* framework comprises three modules: the CLIP-MMFE module, the GASP-based LLM module, and the planning module. Since each module is trained independently, we discuss the training details for each module separately, beginning with the CLIP-MMFE module.

**Details of *CLIP-MMFE* Module**   The network architecture of the satellite encoder ($s_\theta$) used in the CLIP-MMFE module is identical to the image encoder of CLIP ($f_\phi$). As mentioned in Section 4, during the training of the satellite encoder, the CLIP image encoder remains frozen. We utilize a pre-trained CLIP encoder available on Hugging Face [2] to guide the training of the satellite encoder. We choose a learning rate of 1e-4, a batch size of 256, the number of training epochs as 300, and the Adam optimizer to train the parameters $\theta$ of the satellite encoder. The objective function we use to train the satellite encoder is defined in equation 2.

**Details of GASP-based LLM Module**   Next, we provide details of the GASP-based LLM module. We select Falcon, a popular LLM architecture, for our GASP-based LLM module. We initialize the GASP-based LLM module using a pre-trained Falcon model available on Hugging Face [3]. We attach a multi-modal projection layer to the output of CLIP-MMFE, which transforms the image representations into LLM representation space. Additionally, apart from the <GOAL> token, we add relative position encoding to each state. This helps the GASP module to learn the relative orientations and positions of aerial images it has seen previously. Note that, relative positions are measured with respect to the top-left position of the search area. Performance comparisons with different LLM architectures are discussed in Section I. The loss function we use to optimize the GASP-based LLM module is defined in equation 3. We use a learning rate of 1e-4, batch size of 1, number of training epochs as 300, and the Adam optimizer to train the parameters of the LLM module.

---

[2]https://huggingface.co/openai/clip-vit-base-patch16
[3]https://huggingface.co/tiiuae/falcon-7b

**Planning module**    Finally, the planner module consists of an actor and a critic network. Both these networks are simple MLPs, each comprising three hidden layers with Tanh non-linear activation layers in between. We also incorporate a softmax activation at the final layer of the actor network to output a probability distribution over the actions. We use a learning rate of 1e-4, batch size of 1, number of training epochs as 300, and the Adam optimizer to train both the actor and critic network. We use the loss function as defined in equation 4 to optimize both the actor and critic network. We choose the values of $\alpha$ and $\beta$ (as defined in equation 4) to be 0.5 and 0.01 respectively. We also choose the clipping ratio ($\epsilon$) to be 0.2. We select discount factor $\gamma$ to be 0.99 for all the experiments and copy the parameters of $\pi$ onto $\pi_{old}$ after every 4 epochs of policy training. Note that in this work we consider an action invalid if it causes an agent to move outside the predefined search area. During training, we divide the images into $5 \times 5$ non-overlapping pixel grids each of size $300 \times 300$.

**Compute Resources**    We use a single NVidia A100 GPU server with a memory of 80 GB for training and a single NVidia V100 GPU server with a memory of 32 GB for running the inference. It required approximately 24 hours to train our model for about 300 epochs with 830 AGL tasks each with a maximum exploration budget ($\mathcal{B}$) of 10, while inference time is 24.629 seconds for 890 AGL tasks with $\mathcal{C} = 6$ on a single NVidia V100 GPU.

## L    MM-GAG Dataset Details

The dataset was built by collecting high-quality geo-tagged images from smartphone devices. After filtering the images, the dataset contained 73 images in total across the globe. The global coverage of the dataset is depicted in Figure 16. For each ground-level image, we downloaded high-resolution satellite image patches at a spatial resolution of 0.6m per pixel. To be able to create 5x5 grids, we downloaded 10x10 satellite image patches of size 256x256 pixels. Further, we automatically captioned each ground-level image using LLaVA-7b [16] [4], a multimodal large language model. We used the prompt: *"Describe the contents of the image in detail. Be to the point and do not talk about the weather and the sky"*.

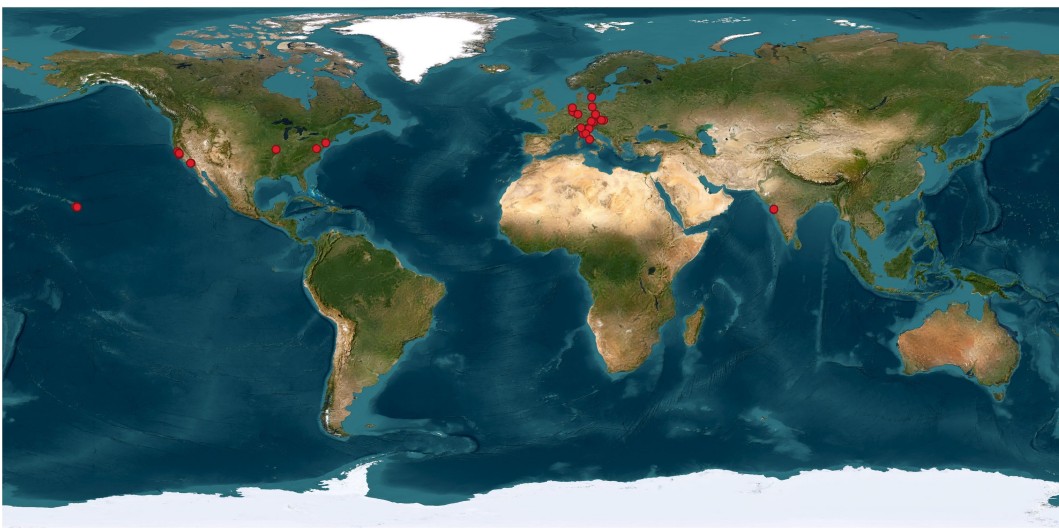

Figure 16: Locations of the samples in our MM-GAG dataset.

**Sampling locations.** The ground-level images were collected from a diverse group of users via a small-scale crowdsourcing effort. We made every effort to ensure the images were sourced from a wide range of countries (the dataset covers 11 countries). The sampling locations were determined by the GPS information embedded in the EXIF data of the images, not by manual selection. The purpose of using the privately sourced images was to avoid leakage into any of the foundation models.

---

[4]https://huggingface.co/liuhaotian/llava-v1.5-7b

**Data filtering.** Initially, we collected 82 images. We applied a basic filter to the collected ground-level images, based on the availability of GPS data. Since we needed to retrieve satellite imagery corresponding to each ground-level image, we required that each image include GPS information in its EXIF data. Images lacking GPS information were excluded.[5] We did not apply any further filtering. Finally, our dataset comprises 73 ground-level images.

**Regarding potential biases.** As mentioned before, we did our best to ensure diversity among the ground-level images. Our dataset features both indoor and outdoor scenes from 11 different countries. Furthermore, we report in Table 16 the average pairwise similarity between the images in our dataset, computed using cosine similarity of the corresponding image embeddings from various vision models.

Table 16: Comparison of average pairwise similarity between the images in our dataset, computed using different backbones.

| Vision Model | DinoV2 [22] | SigLIP [44] | CLIP [28] |
|---|---|---|---|
| Avg. Pairwise Similarity | 0.10±0.22 | 0.32±0.17 | 0.56±0.13 |

The low average pairwise similarity suggests that the images in our dataset represent a diverse range of concepts. Finally, we would like to emphasize that a single ground-level image can be utilized to generate up to 300 potential start and goal scenarios by spatially adjusting the grid of satellite images. By randomly initializing start and goal locations and averaging the results over 5 different random seeds enabled us to robustly evaluate our model.

**Link to the dataset.** Our dataset is publicly available at this link.

## M   Analyzing Geo-Localization Performance across Multiple Trials

Here, we compare the search performance of all the baseline approaches along with *GOMAA-Geo* across 5 different trials for different evaluation settings using a boxplot. In Figure 17 and 18, we report the result with $\mathcal{B} = 10$ and $5 \times 5$ grid configuration using Masa and MM-GAG dataset respectively.

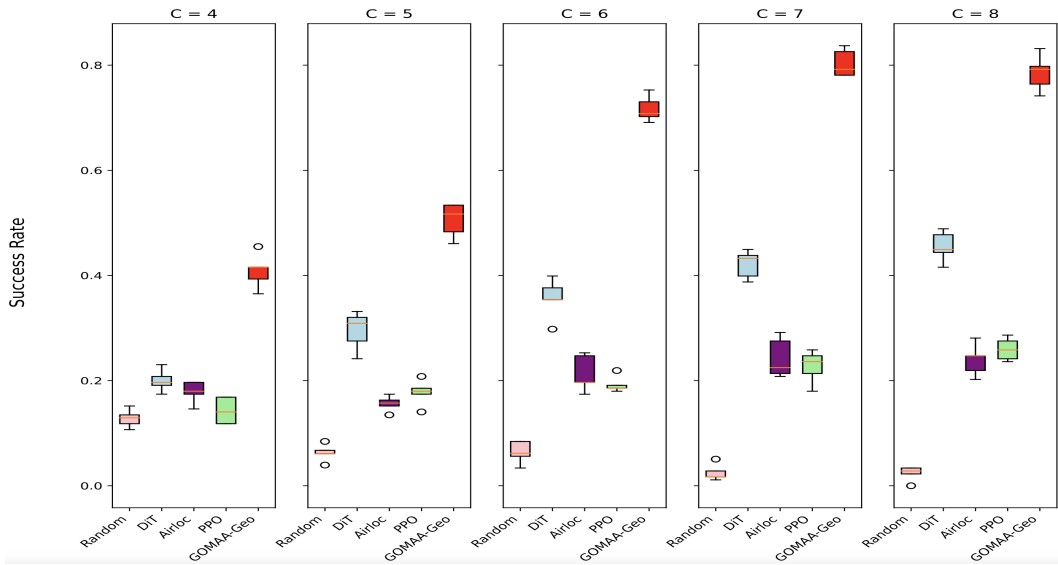

Figure 17: Boxplot of different methods for different choices of $\mathcal{C}$ using Masa Dataset.

---

[5]Note that no GPS information was provided to *GOMAA-Geo* when tackling the active geo-localization task (as we are interested in tackling scenarios where GPS information is lacking or disturbed), i.e. the GPS positions were only used during the dataset creation, to match ground-level views to satellite imagery.

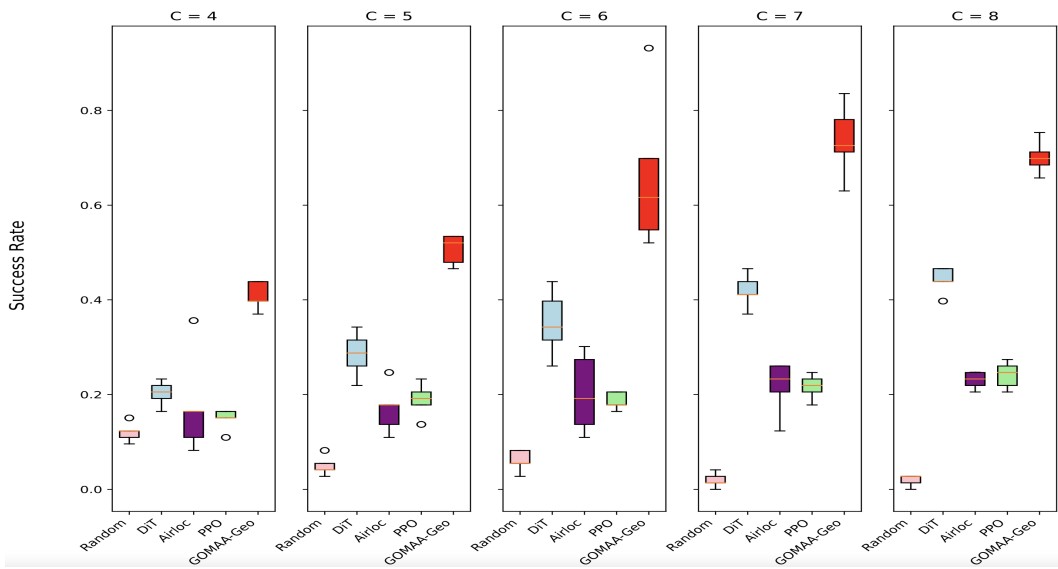

Figure 18: Boxplot of different methods for different choices of $\mathcal{C}$ using MM-GAG dataset.

## N    Limitations

As a work that pioneers the task of modality-invariant active geo-localization, some limitations have been included in order to provide a streamlined, reproducible and controllable first test and experimentation setup. Such limitations include: **(i)** the ego-pose of the agent is assumed to always be known and noise-free (whereas in real-world deployment, a UAV would need to handle this as well); **(ii)** we have utilized a grid-like state-action space as opposed to a continuous one (which would be the case in real UAVs); and **(iii)** the agent's movements are constrained to a plane, i.e. it cannot change elevation during exploration (however, even in real-world applications, there may be circumstances where a UAV would have to hover at a limited and constant elevation, e.g. to avoid heavy winds during a search-and-rescue operation in a windy area). We emphasize that these limitations are relatively minor when considered in relation to the several novel aspects introduced in our work – and thus we leave further steps towards real-world deployment for future work.

## O    Broader Impacts

The development of *GOMAA-Geo* has the potential to create significant positive downstream impacts across a wide range of applications, most notably in enhancing automated search-and-rescue operations and environmental monitoring. These advancements promise to deliver tangible benefits in terms of efficiency, accuracy, and overall effectiveness, fundamentally transforming how these critical tasks are performed.

**Search-and-rescue operations.** *GOMAA-Geo* has potential to revolutionize search-and-rescue missions by enabling flexible, rapid and precise localization of individuals in distress. In disaster-stricken areas, such as those affected by earthquakes, floods, or wildfires, the ability of *GOMAA-Geo* to process various forms of data (e.g., aerial images, ground-level photographs, or textual descriptions) ensures that rescue teams can quickly locate and assist survivors. This can lead to a significant reduction in the time taken to find missing persons, ultimately saving lives and resources.

**Environmental monitoring.** Environmental monitoring stands to benefit immensely from the capabilities of *GOMAA-Geo*. By efficiently analyzing and interpreting data from diverse modalities, this technology can be used to enhance the monitoring of wildlife, forests, water bodies, and other ecological assets. This improved monitoring capability can lead to better-informed conservation efforts, more effective management of natural resources, and early detection of environmental hazards, contributing to the preservation of ecosystems and biodiversity.

**Further considerations.** While the potential benefits of *GOMAA-Geo* are substantial, it is imperative to consider the ethical implications of its deployment. We strongly advocate for the responsible use of this technology, emphasizing applications that contribute to the common good. Specifically, we advise against and condemn any misuse of *GOMAA-Geo* in contexts that could harm individuals or societies, such as in warfare or surveillance that infringes on privacy rights.

## P  Additional Visualizations of Exploration Behaviors of *GOMAA-Geo* in Scenes Containing Object Categories Other than Forests and Buildings

In Fig. 19 we present further visualizations of *GOMAA-Geo*'s exploration behavior in scenes that contain additional land cover types (i.e., beyond buildings and forests). For example, the leftmost figure depicts a visual scene with a *lake*, with the goal located in the middle of the lake. The middle figure also features a *lake*, with the goal situated near the shore. In both of these scenarios, we observe that the *GOMAA-Geo* agent (red line) successfully locates the goal and follows the optimal path. Finally, the rightmost figure shows a scene with a large *parking lot*, where the goal is situated. Even in this scenario, the *GOMAA-Geo* agent successfully locates the goal in a minimal number of steps. The exploration behaviors presented in the main paper, appendix, and Fig. 19 demonstrate *GOMAA-Geo*'s zero-shot generalization capability for active geo-localization across diverse scenes.

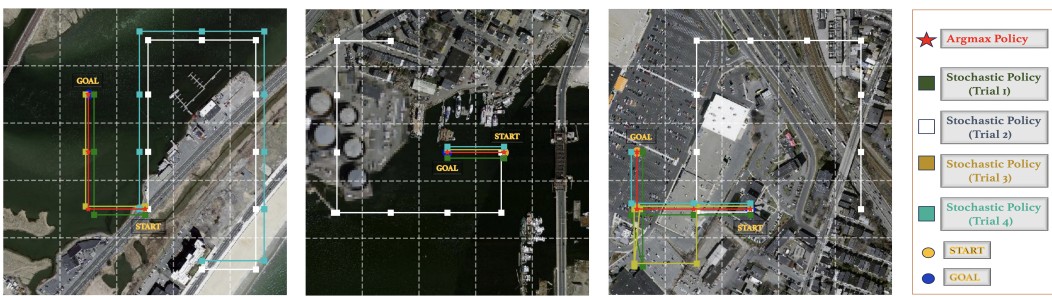

Figure 19: Exploration behaviors of *GOMAA-Geo* across different scenes.

## Q  Exploration Behavior of *GOMAA-Geo* with Goal not at a Boundary of the Search Area

In Figure 20 we present additional visualizations of *GOMAA-Geo*'s exploration behavior in scenes where the goal is not at a boundary. Similar to scenarios with the goal at the boundary, we observe that *GOMAA-Geo* successfully locates the goal by following an optimal path in these cases as well. These visualizations, along with those presented in the main paper and appendix, demonstrate that *GOMAA-Geo* is efficient in locating the goal, regardless of its position in the search area.

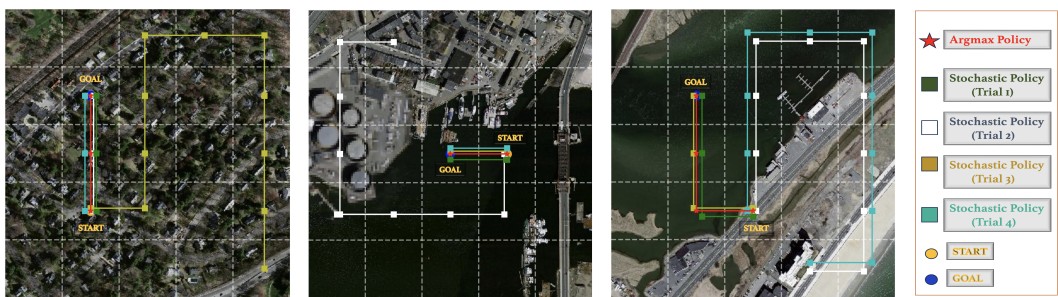

Figure 20: Example exploration behaviors of *GOMAA-Geo* with goals not at a border.

