# OpenReview forum: "GOMAA-Geo: GOal Modality Agnostic Active Geo-localization"
_NeurIPS.cc/2024/Conference — NeurIPS 2024 poster_

### Official Review · Reviewer_DjP3 · 2024-06-24

**Soundness:** 4
**Presentation:** 3
**Contribution:** 3
**Rating:** 5
**Confidence:** 4

**Summary:**

The paper introduces GOMAA-Geo, a system designed for active geo-localization tasks where the goal can be specified in various modalities, including text, ground-level, or aerial imagery. The framework uses a combination of cross-modality contrastive learning to align representations across different modalities, supervised foundation model pretraining, and reinforcement learning to derive effective navigation and localization policies. It demonstrates strong performance in both same-modality and cross-modality localization tasks, with an emphasis on zero-shot generalization to new, unseen datasets.

**Strengths:**

1. Modality Agnosticism: The ability to handle goal specifications in various modalities makes GOMAA-Geo highly versatile and applicable in diverse real-world scenarios, particularly in search-and-rescue operations where goal modalities can vary significantly.

2. Zero-Shot Generalization: The framework is effective at generalizing to completely new environments and goal modalities it has never seen during training, which is crucial for practical applications where pre-collected data might not be available.

3. Robust Performance: Extensive testing shows that GOMAA-Geo consistently outperforms other state-of-the-art methods in terms of success rates across multiple datasets and modalities, confirming its efficacy.

**Weaknesses:**

1. Complex Integration: The integration of multiple complex components—contrastive learning, supervised pretraining, and reinforcement learning—while effective, can be challenging to optimize and maintain, potentially limiting its adaptability.

2. Computational Demands: The processes involved, particularly the training across different modalities and the reinforcement learning steps, might require substantial computational resources, which could be a barrier to deployment in resource-constrained environments.

3. Limited Discussion on Failure Cases: The paper could benefit from a more detailed discussion on scenarios where GOMAA-Geo fails or performs suboptimally, which would be valuable for future improvements and practical deployments.

**Questions:**

N/A

**Limitations:**

Yes.

---

> ### Author Rebuttal · Authors · 2024-08-06
>
> Thanks, we appreciate the reviewer's valuable feedback!
>
>
> > **Q1**: Complex Integration: The integration of multiple complex components—contrastive learning, supervised pretraining, and reinforcement learning—while effective, can be challenging to optimize and maintain, potentially limiting its adaptability.
>
> **A1**: We agree that our approach relies on more than a single “end-to-end learning protocol”, and in that sense, it might be somewhat more laborious to train our method from scratch. However, a similar argument can be made for how the 3-step approach (contrastive learning; supervised pretraining; reinforcement learning) makes the overall system more robust, modular, and adaptable. By this, we mean that, assuming e.g. a SOTA reinforcement learning algorithm emerges in the future, then most of the system can be kept intact and only the third step needs to be replaced with that new RL approach. Finally, note that our work pioneers a new task setup and set of approaches for tackling it, and future work may well result in even more streamlined approaches for the task (which we encourage also via releasing all code and pre-trained models upon paper acceptance).
>
> > **Q2**: Computational Demands: The processes involved, particularly the training across different modalities and the reinforcement learning steps, might require substantial computational resources, which could be a barrier to deployment in resource-constrained environments.
>
> **A2**:  We first want to re-emphasize that our GOMAA-Geo model is _**not**_ trained across different modalities. This is also described e.g., in the abstract (Line 20-21), Section 3 “Active Geo-localization Setup” (Line 137-138), and Section 5 “Experiments and Results” (Line 271-272). We do agree, however, that RL training is typically a time- and/or resource-consuming task, which may indeed be difficult in resource-constrained environments. But, we would also like to highlight here that all models were trained on a single dual GPU workstation, which we believe is well within the budget of many research labs. More on compute resources/requirements are described around Lines 756-760 in the appendix.  Finally, as also listed by you under the strength “2. Zero-Shot Generalization”, our results clearly show that GOMAA-Geo generalizes well to completely new environments and goal modalities, so e.g. research groups/practitioners with less compute available may still be able to successfully use these approaches without further RL training.
>
> > **Q3**: Limited Discussion on Failure Cases: The paper could benefit from a more detailed discussion on scenarios where GOMAA-Geo fails or performs suboptimally, which would be valuable for future improvements and practical deployments.
>
> **A3**: Thank you for pointing this out; we will add several failure case examples with associated discussions to the appendix of the revised version. Please see also the **attached PDF** with examples of failure cases, where one can see that in scenarios where the goal patch is identical to many other patches in the search space, our GOMAA-Geo agent can become unsuccessful (but note that it would be confusing even for humans in these settings, as we also rely on more discriminative aspects when looking for something).

---

> > ### Comment · Reviewer_DjP3 · 2024-08-11
> >
> > Thanks for the response, I have reviewed all the other reviewers' comments and the authors' responses. I agree with Reviewer 9JWt that the main contribution of the dataset lacks some details and the zero-shot performance requires a more reasonable explanation. While I also acknowledge the paper's significant contribution to the community, I would like to change my rating to borderline accept.

---

> > > ### Author Response · Authors · 2024-08-11
> > >
> > > Thank you for the comments and your engagement. We provide our response to your comments below:
> > >
> > > > **Q1**: "Lack some details regarding the dataset"
> > >
> > > **A1**:
> > >
> > > Thank you for your comment. We refer you to our detailed response ([link](https://openreview.net/forum?id=gPCesxD4B4&noteId=wpmhB8kN3m)) to Reviewer 9JwT's concerns about the dataset. We believe we have thoroughly addressed all relevant details as requested. Should you have any further questions or concerns regarding the dataset, please do not hesitate to reach out to us. The dataset has already been made available at this anonymous [link](https://huggingface.co/datasets/Gomaa-Geo/MM-GAG/tree/main).
> > >
> > > Finally, as you mentioned in your response "lack some details", **it would be beneficial if you could specify which specific details you are referring to. This will enable us to respond more effectively.** Next, we clarify the reason behind the superior zero-shot generalization performance of our model.
> > >
> > > > **Q2**: Reason for superior zero-shot generalization performance?
> > >
> > > **A2**: The CLIP-MMFE module utilizes a satellite image encoder to project satellite images into the CLIP feature space. The original CLIP model, trained on 400 million image-text pairs, predominantly consists of diverse ground-level images. This extensive training allows the CLIP vision encoder to develop robust, generalized visual representations, making it a powerful feature extractor with strong zero-shot capabilities.
> > >
> > > The satellite image encoder of the CLIP-MMFE module, although trained on a more specialized dataset of satellite images, maps these images into the CLIP feature space. As a result, it inherits the CLIP model’s exceptional zero-shot performance. This inheritance occurs because the satellite encoder benefits from the rich and diverse visual features learned by the CLIP vision encoder. By mapping satellite images to the same feature space as the CLIP model, the satellite encoder leverages the extensive training of the CLIP model, thereby achieving improved zero-shot generalization capabilities.
> > >
> > > In essence, the effective feature extraction and zero-shot learning abilities of the CLIP vision encoder are extended to the satellite image encoder and is achieved by aligning the feature space of the satellite image encoder of the CLIP-MMFE module and CLIP vision encoder as discussed in lines 149-170, enhancing its performance in recognizing and interpreting new or unseen satellite imagery within the well-established CLIP feature space.
> > >
> > > To substantiate our argument, we have conducted numerous experiments and provided both qualitative and quantitative analyses throughout the paper. We would like to emphasize one key result here.
> > >
> > > To validate how aligning features from different modalities to the unified CLIP space—achieved through the CLIP-MMFE module—enhances zero-shot performance, we conducted an experiment (**detailed in Appendix D**) comparing it to a modality-specific feature extractor, such as SatMAE, a foundational satellite image encoder. Although the performance of both approaches is comparable when the goal modality is satellite imagery, a significant performance gap emerges in zero-shot evaluations with other goal modalities. This highlights the benefit of aligning features from different modalities to the robust, zero-shot generalizable CLIP space, as demonstrated by our results.
> > >
> > > We would be very happy to provide further clarification and hopefully clarify your concerns (if any) about zero-shot generalizability before the discussion period concludes.

---

> ### Author Response · Authors · 2024-08-12
> **Follow-up to our response**
>
> Dear Reviewer DjP3,
>
> In our rebuttal, we have already addressed all your previous comments and have answered 9JwT's queries about our dataset, and provided all necessary details along with a link to the dataset.
>
> As we approach the end of the discussion period, we wanted to follow up to see if our responses have addressed your concerns. We would be very grateful to hear additional feedback from you and will provide further clarification if needed. If you believe our response has adequately resolved the issues you raised, we kindly ask you to consider the possibility of raising the score.
>
> Thank you again for your time and effort.
>
>
> Thanks,
>
> Authors

---

> > ### Comment · Reviewer_DjP3 · 2024-08-13
> >
> > Thanks for these further responses, and I have no other questions. I hold a positive rating of 5, and I lean to accept this paper.

---

### Official Review · Reviewer_noHa · 2024-07-11

**Soundness:** 2
**Presentation:** 2
**Contribution:** 2
**Rating:** 4
**Confidence:** 4

**Summary:**

In this paper, the author proposes a direction classification task, which is for the drone path navigation.
My main concerns are about the task.
Usually we have the global satellite-view image, shall we estimate the direction?
Most works do the location retrieval, which is a common practise in the geolocation.

**Strengths:**

In this paper, the author proposes a direction classification task, which is for the drone path navigation.
The method seems sound.
The idea is presented clear.

**Weaknesses:**

1. My main concerns are about the task.
Usually we have the global satellite-view image, shall we estimate the direction?
Most works do the location retrieval, which is a common practise in the geolocation.

2. Can the network general to unseen scenarios?
For example, there are many similar buildings or similar forests.

3. I am not sure the network could predict the future according to the history.
Because there are no global map as input.

4. The drone usually do not go back, so the estimation space is only 3 directions.
If we further consider the boundary or corner, there are only 2 / 1 direction. Will it affect the model?

5. I see the samples shown in the paper.
I see a biases toward the boundary and corner. Could you show more samples with destination at the center?

6. Missing alignment.
How about give a destination between the two patches?

7. The direction estimation is not new. Similar methods have been explored in the RL and detection.

8. Typos. i.e. should be i.e.,

**Questions:**

Please see the Weakness.

**Limitations:**

Please see the Weakness.

---

> ### Author Rebuttal · Authors · 2024-08-06
>
> Thank you for reviewing our paper! We hopefully address all of your current concerns / potential misunderstandings below.
>
>
> > **Q1**: My main concerns are about the task. Usually we have the global satellite-view image, shall we estimate the direction?
>
> **A1**: This appears to be a misunderstanding. We are **_not_** proposing a “direction classification task” (DCT), i.e., the task is not merely to predict a direction pointing toward a specific goal, but to _explore a partially explored environment to reach the goal_. This may imply that the agent needs to also explore places that are not taking it closer to the goal, to reveal contextual information which may later be useful for efficiently moving towards the goal. We denote this novel task as _goal modality agnostic active geo-localization_.
>
> A DCT is _only used as a pre-training task_ (see Sec. 4, L. 171-210, Fig. 2). After pre-training, our full GOMAA-Geo system is refined with RL (see Sec. 4, L. 211-243). In Table 5 we clearly show the benefit of the additional RL-based fine-tuning on top of the DCT warm-up task.
>
> **Comparison to other works:** There indeed exist many works on what we call _static_ geo-localization, which often assume access to a broader satellite view perspective, and where the mapping between a ground-level and top-view (satellite) image is treated as an image retrieval problem. We discuss this in Related Work (see e.g. L. 69-75). However, high-resolution satellite imagery is not available publicly everywhere in the world, and GPS information can sometimes be absent or unreliable (e.g. in search-and-rescue operations after natural disasters or warfare activity). Motivated by this, our work tackles geo-localization in partially observable GPS-denied environments, and we specifically tackle it in a _goal modality agnostic_ setup, since if GPS is not available, an agent must be able to robustly find a given location regardless of how it is specified.
>
> > **Q2**: Can the network generalize to unseen scenarios? For example, there are many similar buildings or similar forests.
>
> **A2**: Absolutely! This is one of the key strengths of our framework, which we validated e.g. by conducting extensive experiments with open-source datasets specifically designed to test GOMAA-Geo's ability to handle a range of unseen pre- and post-disaster real-world scenarios. Despite not being trained on any post-disaster aerial images, GOMAA-Geo performs well on images captured after disasters (Table 3). See also L. 307-321 for a detailed discussion on this.
>
> Additionally, the experimental section and the appendix contain extensive experimental results regarding generalization to unseen scenarios and goal modalities. See e.g. L. 287-306, Table 2, Appendix B, C, F. See also the **attached PDF** for example exploration behaviors of GOMAA-Geo in scenes containing additional land cover types (i.e., beyond buildings and forests). These will be added to the appendix of the revised version.
>
> > **Q3**: Not sure the network could predict the future according to the history.
>
> **A3**: The agent is _not_ tasked to predict the future based on its history of sequentially observed top-view glimpses and actions, but rather to predict a promising next action that ultimately leads the agent to reach its goal. We discuss this in the main paper (L. 376-387) and show clearly in Fig. 6 and in Appendix F that the full history is useful for this, as opposed to e.g. only basing the decision on the most recent state.
>
> > **Q4**: "The drone usually do not go back, so the estimation space is only 3 directions. If we further consider the boundary or corner, there are only 2 / 1 direction. Will it affect the model?"
>
> **A4**: When in a non-boundary position, the agent can always select any of the 4 actions (up, down, left, right), but as it is aware of its relative position in its search area (_not_ global GPS location; see L. 110) and a history of past relative positions, it quickly learns during training -- based on the reward (eq. (6)) that penalizes such abnormal behaviors -- that it is mostly not beneficial to move to its previous position. When it is on a boundary, we invalidate actions that would take the agent outside the search area (see L. 753-754), which is handled by sampling / selecting only from the available actions. We rarely observed abnormal behaviors from our trained GOMAA-Geo model, as evidenced by numerous visual illustrations of exploration behaviors in different scenarios throughout the paper (see e.g. Fig. 4, 5, 6 in the main paper; also Fig. 7-12 in the Appendix).
>
> > **Q5**: Could you show more samples with destination at the center?
>
> **A5**: See the **attached PDF** with additional visual results, where the goal is in the middle parts of the search area (will also be added to the appendix in the revision). Note that cases with the goal closer to a border typically correspond to scenarios where the start-to-goal distance is larger, so the examples that we focused most on in the paper can be expected to be more difficult.
>
> > **Q6**: Missing alignment. How about give a destination between the two patches?
>
> **A6**: We unfortunately have trouble understanding this remark. It seems you are asking if we could represent goal locations that aren't centered within a given cell. Our current framework assumes a discrete domain and doesn't require that goal locations be centered within the target cell, only that they are contained within the field of view.
>
> > **Q7**: The direction estimation is not new. Similar methods have been explored in the RL and detection.
>
> **A7**: This misunderstanding was addressed in A1 above. We emphasize that *we are not proposing direction estimation*. Rather, we propose a novel, more complex task of goal modality agnostic active geo-localization. To the best of our knowledge, this task has not been explored by others and is therefore novel.
>
> > **Q8**: Typos.
>
> **A8**: Thanks, we'll fix the identified typos in the revision.

---

> ### Author Response · Authors · 2024-08-12
> **Follow-up to our response**
>
> Dear Reviewer noHa,
>
> In our rebuttal, we addressed all your concerns in detail including your concern regarding the task, and also provided additional visualizations that you requested. We will include all the additional visualizations in our revised draft.
>
> As we approach the end of the discussion period, we would like to follow up to ensure that our responses have satisfactorily addressed your concerns. We would greatly appreciate any further feedback you may have and are prepared to offer additional clarification if necessary. If you believe our response has adequately resolved the issues you raised, we kindly ask you to consider the possibility of raising the score.
>
> Thank you again for your time and effort.
>
>
> Thanks,
>
> Authors

---

### Official Review · Reviewer_9JwT · 2024-07-13

**Soundness:** 2
**Presentation:** 2
**Contribution:** 3
**Rating:** 5
**Confidence:** 3

**Summary:**

This paper introduces GOMAA-Geo, a framework designed to enhance active geo-localization for the UAV, enabling them to find targets specified through various modalities like natural language or imagery. It addresses the challenge of efficient localization in dynamic environments without specific training on disaster scenarios. Utilizing a modality-agnostic approach with cross-modality contrastive learning and a combination of pretraining and reinforcement learning, GOMAA-Geo achieves robust navigation and localization. It conducted extensive experiments across various datasets and demonstrated its effectiveness.

**Strengths:**

- It proposes a new dataset named Multi-Modal Goal Dataset for Active Geolocalization (MM-GAG) for active geo-localization across different goal modalities: aerial images, ground-level images, and natural language text.
- This paper introduced a framework named GOMAA-Geo, which utilizes a modality-agnostic approach, combining cross-modality contrastive learning to generalize across different goal specifications and modalities.
- Extensive experiments are done in this paper, which is quite laborious.

**Weaknesses:**

- The introduced dataset is the main contribution of the paper, but the details of the dataset are very limited, and there is no more information in the appendix, such as the details of data collection, why the zero-shot generalization performance of the evaluation model is better?
- The writing needs to be improved, especially the method part, which is quite confusing. In 240 lines, it is claimed that the complete GOMAA-Geo framework integrates all the components introduced before, but it is not easy to find the corresponding relationship from the text to the figure, and the connection between Figure 1, Figure 2, and Figure 3 is also difficult to sort out.

**Questions:**

What is the full name of the UAV? There is no explanation in the whole paper, which will cause confusion to readers.

**Limitations:**

It addresses the limitations in the Appendix.

---

> ### Author Rebuttal · Authors · 2024-08-06
>
> While we very much appreciate the feedback on our work, we were surprised to find that this reviewer has recommended rejecting the paper, given that the listed "Strengths" seem to clearly outweigh concerns listed under "Weaknesses" (where no technical concerns about our work were mentioned) nor under "Questions" (where only a clarification about an abbreviation is requested). We address the listed concerns and the question below.
>
> > **Q1**: The introduced dataset is the main contribution of the paper.
>
> **A1**: We wish to emphasize that while our dataset is indeed a significant contribution, we do not view it as the *main* contribution. Rather, our main contribution is the introduction and extensive evaluation of a novel framework designed to effectively address goal modality agnostic active geo-localization, even when the policy is trained solely on data from a single goal modality. However, we do agree that our dataset is also an important contribution, both on its own (we carefully considered how to design it so that it is easy to use for future researchers, and as mentioned will release it together with the code upon paper acceptance), and because it allows for assessing and evaluating various methods within our novel task setting (i.e., it is key for validating our main contribution). We summarize our contributions in Lines 55-67.
>
> > **Q2**: The details of the dataset are very limited, and there is no more information in the appendix, such as the details of data collection.
>
> **A2**: We describe the proposed MM-GAG dataset on Lines 263-276 in Section 5 “Experiments and Results”, and provide additional information in the Appendix section titled “MM-GAG Dataset Details”. Together, these parts of the paper describe key dataset details such as (i) locations in the world where the data were captured (Fig. 15); (ii) **details of data collection** / creation procedure (beginning at Line 762 in the appendix); (iii) number of evaluation scenarios covered (Line 273-275). If the reviewer can identify concrete missing details, we would be happy to provide these as well. And, of course, we will release the dataset for future research in this field upon acceptance on GitHub along with detailed documentation. We plan to use the GitHub issue tracker to gather requests for additional information about the dataset to ensure ease of use for future researchers.
>
> > **Q3**: Why the zero-shot generalization performance of the evaluation model is better?
>
> **A3**: We attribute the model's superior zero-shot generalization performance to the CLIP-MMFE module (see Lines 149 - 170 for a detailed discussion on this; we also encourage the reviewer to see Lines 323- 331 where we empirically validate the effectiveness of CLIP-MMFE and provide corresponding visualizations in Fig. 4). One of the most impressive features of the CLIP-MMFE module is its ability to align features across different modalities (such as aerial imagery and text) with the CLIP ground-level image encoder known for its zero-shot generalization capabilities.
>
> Moreover, based on the results in Table 2, one of the unseen modalities during training ("Ground Image") appears surprisingly to lead to better results compared to the modality seen during training ("Aerial Image"). The reason for this is that the ground-level image task is a little bit easier because it is constrained to be in locations with discriminative information (the available ground-level images typically depict discriminative scenery such as unusual/salient buildings etc), whereas aerial images are not similarly constrained. The way in which we see that "Ground Image" gives slightly better results is by computing the average across the different C-values (i.e. across columns) of Table 2 and getting the mean results for Text, Ground Image, and Aerial Image to be 0.6008, 0.6145 and 0.6034, respectively. Thus the results are on average best for Ground Image and worst for Text, with Aerial Image in between. Note, however, that all results are quite similar, which showcases the robustness across the various modalities. We also refer the reviewer to Appendix D, where we discuss this “trade-off” between modality-specific vs. modality-agnostic goal representation in active geo-localization. Our findings suggest that on the one hand, modality-agnostic representations are beneficial for addressing active geo-localization problems across diverse goal modalities, and on the other hand, they are equally competitive with models that are designed to solve modality-specific active geo-localization tasks, such as SatMAE-Geo.
>
>
> > **Q4**: The writing needs to be improved, especially the method part, which is quite confusing. In 240 lines, it is claimed that the complete GOMAA-Geo framework integrates all the components introduced before, but it is not easy to find the corresponding relationship from the text to the figure, and the connection between Figure 1, Figure 2, and Figure 3 is also difficult to sort out.
>
>
> **A4**: We appreciate the reviewer's suggestions on improving readability, including improving the consistency between Fig 1-3. We will make improvements as suggested in the revision.
>
>
> > **Q5**: What is the full name of the UAV?
>
> **A5**: UAV stands for “Unmanned Aerial Vehicle”; we will ensure this is clarified in the revised paper.

---

> ### Comment · Reviewer_9JwT · 2024-08-09
>
> The reason why I believe the dataset is the main contribution is twofold: (1) As stated by the author in the rebuttal, "We summarize our contributions in Lines 55-67," for the contribution paragraph, the first part introduces the GOMAA-Geo framework, and the second part discusses the dataset contribution: "We create a novel dataset to assess various approaches for active geo-localization across three different goal modalities: aerial images, ground-level images, and natural language text." (2) In this paper, the authors claim the proposed open-source dataset MM-GAG is "currently available for evaluating the zero-shot generalizability of GOMAA-Geo across diverse goal modalities, such as ground-level images and natural language text." (Lines 266-270). They conduct a variety of experiments using MM-GAG, detailed in Table 1, Table 2, and further explored in Table 12, Table 14, and Figure 17 in the Appendix. This indicates that a significant portion of the main experiments and ablations in the paper utilized this dataset, and thus, the quality of the dataset impacts the evaluation results. If the dataset quality is poor, such as having biases, how can I trust that the method proposed by the paper is effective? This is why I am hesitant to give a high score from the dataset perspective. If the author does not consider the dataset as an important contribution, I will reduce my evaluation of the experiments related to MM-GAG, and if evaluating purely from a methodological perspective, I am willing to increase my score.
>
> Additionally, regarding the details of the data, although more details are provided in the supplementary materials from Lines 762-769, I still hope for more specific information to validate the dataset's collection rationality. Lines 762-763 mention: "After filtering the images, the dataset contained 73 images in total across the globe." I would like to know how much data was initially collected, what the filtering was based on, what confirms the global coverage of the dataset, why these specific locations were chosen for sampling, and how the current dataset ensures that it is not biased.

---

> > ### Author Response · Authors · 2024-08-10
> > **Additional Details Regarding the Dataset**
> >
> > Thank you for your comment. We’re pleased to provide additional details about the dataset that you requested. Please see our detailed response below.
> >
> > **Regarding sampling locations**: The ground-level images were collected from a diverse group of users via a small-scale crowdsourcing effort. We made every effort to ensure the images were sourced from a wide range of countries. The sampling locations were determined by the GPS information embedded in the EXIF data of the images, not by manual selection. The purpose of using the privately sourced images was to avoid leakage into any of the foundation models.
> >
> > **Regarding filtering**: Initially, we collected 82 images. We applied a basic filter to the collected ground-level images, based on the availability of GPS data. Since we needed to retrieve satellite imagery corresponding to each ground-level image, we required that each image include GPS information in its EXIF data. Images **lacking GPS information** were excluded. We did not apply any further filtering. Finally, our dataset comprises 73 ground-level images.
> >
> > **Regarding biases**: As mentioned before, we did our best to ensure diversity among the ground-level images. Our dataset features both indoor and outdoor scenes from 11 different countries. Furthermore, we report the average pairwise similarity between the images in our dataset, computed using cosine similarity of the corresponding image embeddings from various vision models:
> >
> > |         Vision Model              | DinoV2 [1] | SigLIP [2] | CLIP [3]        |
> > |--------------------------|--------|--------|-------------|
> > | Avg. Pairwise Similarity |   0.10±0.22    |     0.32±0.17   | 0.56±0.13 |
> >
> > The low average pairwise similarity suggests that the images in our dataset represent a diverse range of concepts. Finally, we would like to emphasize that a single ground-level image can be utilized to generate up to 300 potential start and goal scenarios by spatially adjusting the grid of satellite images. By randomly initializing start and goal locations and averaging the results over 5 different random seeds enabled us to robustly evaluate our model (Line 289).
> >
> > **Link to the dataset**: An anonymous link to the dataset was included in the supplementary material of our initial submission. For your convenience, here is the [link](https://huggingface.co/datasets/Gomaa-Geo/MM-GAG/tree/main) for further reference.
> >
> >
> > **References:**
> >
> > [1]: Oquab, Maxime, et al. "Dinov2: Learning robust visual features without supervision." Transactions on Machine Learning Research (TMLR), 2023.
> >
> > [2]: Zhai et al. "Sigmoid loss for language image pre-training". International Conference on Computer Vision (ICCV), 2023.
> >
> > [3]: Radford et al. "Learning Transferable Visual Models from Natural Language Supervision". International Conference on Machine Learning (ICML), 2021.

---

> ### Author Response · Authors · 2024-08-12
> **Follow-up to our response**
>
> Dear Reviewer 9JwT,
>
> In our rebuttal, as you requested, we provided all the necessary details about the dataset along with the anonymous dataset link.
>
> As we approach the end of the discussion period, we wanted to follow up to see if our responses address your concerns. We would be very grateful to hear additional feedback from you and will provide further clarification if needed. If you believe our response has adequately resolved the issues you raised, we kindly ask you to consider the possibility of raising the score.
>
> Thank you again for your time and effort.
>
> Thanks,
>
> Authors

---

> > ### Comment · Reviewer_9JwT · 2024-08-14
> >
> > Thanks for your response. I'd like to raise the rating to 5.

---

### Official Review · Reviewer_4muN · 2024-07-13

**Soundness:** 3
**Presentation:** 4
**Contribution:** 3
**Rating:** 6
**Confidence:** 4

**Summary:**

The paper introduces GOMAA-Geo, a novel framework for active geo-localization (AGL) that is capable of zero-shot generalization across different goal modalities. GOMAA-Geo is designed to assist agents, such as UAVs in search-and-rescue operations, to locate targets specified through various modalities (e.g., natural language, ground-level images, or aerial images) using a sequence of visual cues observed during aerial navigation. The framework addresses two main challenges: 1. Dealing with goal specifications in multiple modalities while relying on aerial imagery as search cues. 2. Limited localization time due to constraints like battery life, necessitating efficient goal localization. GOMAA-Geo integrates cross-modality contrastive learning to align representations across modalities, foundation model pretraining, and reinforcement learning to develop effective navigation and localization policies. The framework has been evaluated extensively and shown to outperform alternative approaches, demonstrating its ability to generalize across datasets and goal modalities without prior exposure during training.

**Strengths:**

1. The paper proposes a novel approach that combines cross-modality learning, foundation model pretraining, and reinforcement learning for active geo-localization, which is novel to some extent.
2. GOMAA-Geo's ability to generalize across different datasets and modalities without prior training exposure, enhancing its applicability in diverse real-world scenarios. The framework has undergone rigorous testing and comparison with alternative methods, demonstrating its effectiveness through quantitative and qualitative analyses.
3. The authors have created a new dataset to facilitate benchmarking, which is beneficial for future research in the area.

**Weaknesses:**

While the framework shows promise, it may require further validation in real-world conditions with actual UAVs and under various environmental challenges.

**Questions:**

See the weakness

**Limitations:**

I suggest the authors to discuss the negative impacts of proposed method to our society like public safety.

---

> ### Author Rebuttal · Authors · 2024-08-06
>
> Thank you for your comments. We are pleased to see that you found our proposed framework novel and that the new dataset will be valuable for future research in this field. Below, we address all your comments.
>
> > **Q1**:  While the framework shows promise, it may require further validation in real-world conditions with actual UAVs and under various environmental challenges..
>
> **A1**: In this work we have focused on laying the ground work for the novel and real-world relevant task of Global Navigation Satellite System (GNSS)-denied goal localization, which we call *goal modality agnostic active geo-localization*. Evaluations of our main GOMAA-Geo approach, as well as an extensive set of baselines and ablations, are conducted on multiple real-image datasets (as also pointed out in the reviewer's list of the paper's strengths), including under various environmental challenges (see e.g. generalization from pre-disaster to post-disaster scenarios in Table 3). Natural next steps indeed include evaluating these types of methodologies also using real UAVs, as suggested by the reviewer. We plan to explore this in follow-up work, and given that our code and data will be made publicly available, others will be able to explore this as well.
>
>
> > **Q2**: I suggest the authors to discuss the negative impacts of proposed method to our society like public safety.
>
> **A2**: Please refer to “Broader Impacts” in the appendix (page 21), which discusses both positive and negative potential downstream applications. Specifically, under “Further considerations” (Line 804-808), we discuss potential negative downstream applications of these types of methodologies. We are grateful for the reviewer's consideration of these issues, and wholeheartedly agree that it is important to carefully consider both positive and negative potential downstream effects when developing new methodologies.

---

> > ### Author Response · Authors · 2024-08-12
> > **Follow-up to our response**
> >
> > Dear Reviewer 4muN,
> >
> > Thank you very much for your valuable reviews and the time you've invested. As the author-reviewer discussion period is coming to an end, we sincerely want to confirm whether we have addressed your concerns. If you believe our response has adequately resolved the issues you raised, we kindly ask you to consider the possibility of raising the score.
> >
> > Thank you again for your time and effort.
> >
> >
> > Thanks,
> >
> > Authors

---

> ### Comment · Reviewer_4muN · 2024-08-13
> **Response to Authors**
>
> I have no further questions. I still give positive rating. Thank the authors for your rebuttal.

---

### Author Rebuttal · Authors · 2024-08-06

We thank all the reviewers for their valuable feedback and insightful comments. We are glad that the majority of the reviewers find the zero-shot generalizability across goal modalities to be a strong contribution of our work, and agree with the reviewers that extensive experiments have been conducted in order to validate various claims and contributions. We considered all the concerns raised by the reviewers carefully and hopefully clarified all of their queries. Furthermore, we have attached a PDF with additional visualizations as requested by the reviewers.

---

### Decision · Program_Chairs · 2024-09-25

**Decision:**

Accept (poster)

**Comment:**

This paper received mixed reviews, including a Weak Accept, two Borderline Accepts, and a Borderline Reject. Reviewers who lean towards acceptance appreciate the novelty of the dataset and the technical contributions. However, concerns were raised regarding the lack of detailed dataset information, insufficient insights into certain results, and an absence of discussion on failure cases. The rebuttal effectively addressed most of these issues, leading three reviewers to agree on accepting the paper.

The reviewer 'noHa' (Borderline Reject) raised several concerns, primarily around the problem formulation, particularly the shift from a traditional retrieval approach to a path-finding problem. While these concerns were largely addressed in the rebuttal, 'noHa' remained cautious. The AC, however, supports the authors' novel framing of the problem, recognizing it as a fresh perspective that could inspire further exploration in the field, as current geolocalization methods largely focus on image retrieval.